# RETHINKING THE POLYNOMIAL FILTER OF GNNS VIA GRAPH INFORMATION ACTIVATION THEORY

## ABSTRACT

Recently, it has been a hot research topic to design different polynomial filters in graph neural networks (GNNs). Most of the existing GNNs only pay attention to the properties of polynomials when designing the polynomial filter, thus not only bringing additional computational costs but also ignoring embedding the graph structure information into the construction process of the basis. To address these issues, we theoretically prove that any polynomial basis with the same degree has the same expressive ability and the finely designed polynomial basis that only considers the polynomial property can at most bring linear benefit for GNNs. Then, we propose a graph information activation (GIA) theory that provides a new perspective for interpreting polynomial filters and then analyse some popular bases using the GIA theory. Based on the GIA theory and analysis, we design a simple basis by utilizing the graph structure information and further build a simple GNN (i.e. SimpleNet), which can be applied to both homogeneous and non-homogenous graphs. Experiments on real datasets demonstrate that our SimpleNet can achieve better or comparable performance with relatively less running time compared to other state-of-the-art GNNs.

## 1 INTRODUCTION

Graph Neural Networks (GNNs) are effective machine learning models for various graph learning problems (Wu et al., 2022), such as social network analysis (Li & Goldwasser, 2019; Qiu et al., 2018; Tong et al., 2019), drug discovery (Rathi et al., 2019; Jiang, 2021) and traffic forecasting (Bogaerts et al., 2020; Cui et al., 2019; Li et al., 2018). Generally, the layer of GNNs can be represented as a unified form (Balcilar et al., 2021):

$$H^{(l+1)} = \sigma(\sum_s C^{(s)} H^{(l)} W^{(l,s)}), \qquad (1)$$

where $H^{(l)}$ is the output of the $l^{th}$ layer, $\sigma(\cdot)$ is the activation function, $W^{(l,s)}$ is the weight to be learnt, and $C^{(s)}$ is the $s^{th}$ graph convolution support that defines how the node features are propagated to the neighbouring nodes. Currently, most existing GNNs differ from each other in selecting convolution supports $C^{(s)}$, which are usually designed in a polynomial form.

Specifically, many researchers focus on analyzing the properties of polynomial filters and try to find the optimal basis from the existing polynomials to approximate the filter. For example, Cheb-Net (Defferrard et al., 2016) first adopts an orthogonal Chebyshev basis to approximate the filter. BernNet (He et al., 2021) applies a non-negative Bernstein basis to improve the interpretation of GNN. JacobiConv (Wang & Zhang, 2022) utilizes the orthogonal Jacobi basis due to its flexible approximation ability and faster convergence speed. OptBasisGNN (Guo & Wei, 2023) learns the optimal basis from the orthogonal space. However, these GNNs only consider the properties of polynomials themselves and apply many complicated tricks to implement these bases, thus introducing additional computation costs. For example, BernNet (He et al., 2021) has quadratic time complexity related to the degree of the polynomial, JocobiConv (Wang & Zhang, 2022) requires three iterative calculations to obtain the basis, and OptBasisGNN (Guo & Wei, 2023) introduces an additional orthogonalization process to construct an orthogonal matrix. In summary, there is a growing focus on the nature of polynomials themselves in the design process, which not only introduces additional computational costs, but also neglects to utilize the graph structure information in the design process of the basis.

Based on these discussions, we naturally put forward two problems:

**Problem 1:** Considering the nature of the polynomial alone, how much does the fine design of the polynomial filter help to improve performance? Is it necessary to make such a design?

**Problem 2:** Can we design a polynomial filter in conjunction with the graph structure? Can this help build an explainable and more efficient GNN?

In this paper, we attempt to provide an explicit answer to these two problems, and our contributions are mainly threefold.

- To address the first problem, we theoretically prove that any polynomials with the same degree have the same expressive ability and the same global optimal solution. Also, we prove a theorem to claim that the convergence rate of the unified optimization form for GNNs with the gradient descent method is linear, i.e., the finely designed polynomials can at most bring linear benefits for GNNs. Therefore, we claim that it should not be necessary to over-elaborately design the polynomial bases by only considering the polynomial property.
- To address the second problem, we first propose a graph information activation (GIA) theory that provides a new perspective for the interpretation of polynomial filters in GNNs. Then we analyse some popular bases using the the concepts of positive and proper activation defined in GIA theory. Finally, we propose a simple basis with the embedded graph structure information and further build a simple GNN (i.e. SimpleNet).
- Experimental results on the benchmark node classification datasets verify the superiority of our proposed SimpleNet compared with existing GNNs in both accuracy and running time.

## 2 NOTATIONS AND PRELIMINARIES

**Gradient descent method.** For the optimization problem $\min_{z} f(z)$, we can apply the gradient descent method to solve it, namely,

$$z_{t+1} = z_t - \eta g_t, \tag{2}$$

where $g_t$ is the gradient of the loss function and $\eta$ is the step size.

**Convergence rate.** Suppose $\{f(z_n)\}_{n=0}^{\infty}$ is a sequence that converges to $f(z^*)$, where $z^*$ is the optimal solution of $\min_{z} f(z)$. If positive constants $\mu$ and $\alpha$ exist and

$$\lim_{n \to \infty} \frac{|f(z_{n+1}) - f(z^*)|}{|f(z_n) - f(z^*)|^{\alpha}} = \mu, \tag{3}$$

where $\mu < 1$ and we say $\{f(z_n)\}_{n=0}^{\infty}$ converges to $f(z^*)$ at convergence rate $\mu$ with order $\alpha$. When $\alpha = 1$, we say that $\{f(z_n)\}_{n=0}^{\infty}$ converges to $f(z^*)$ at a linear convergence rate $\mu$.

**Graph Neural Networks.** We denote an undirected and unweighted graph $G$ with vertex set $V$ and edge set $E$ as $G = G(V, E)$, whose adjacency matrix is $A$. The symmetric normalized Laplacian matrix of $G$ is defined as $L = I - D^{-1/2}AD^{-1/2}$, where $D$ is the diagonal degree matrix of $A$. Given a graph signal $x \in R^n$, where $n = |V|$ is the number of nodes in the graph, the graph filtering operation can be represented as $\sum_{k=1}^{K} w_k L^k x$, where $w_k$ is the weight. We denote $L = U\Lambda U^{\top}$ as the eigen-decomposition of $L$, where $U$ is the matrix of eigenvectors and $\Lambda = diag[\lambda_1, ..., \lambda_n]$ is the diagonal matrix of eigenvalues.

**Polynomial filters.** Some GNNs design graph signal filters in the Laplacian graph spectrum, and this type of GNN is called a spectral GNN. To avoid the eigenvalue decomposition of the graph Laplacian matrix, spectral GNNs apply various polynomial bases to approximate the Laplacian filter on the frequency domain of graph, which is called frequency polynomial filter. These frequency polynomial filters construct the $C^{(s)}$ term in Eq. (1) in a uniform polynomial form as other GNNs do, which are uniformly called polynomial filters.

## 3 MAIN WORK

In this section, we present our solutions to the aforementioned two problems.

## 3.1 SOLUTIONS TO PROBLEM 1

To address the first problem, we first theoretically analyze the intrinsic consistency and difference of polynomial filters from two aspects, that is, the expression ability of GNN and the convergence rate of the optimization problem underlying the GNN.

### 3.1.1 EXPRESSIVE POWER ANALYSIS OF GNNS

The expressive power of GNNs is the ability to allow GNNs to approximate non-linear mappings from features to labels on the graph. As pointed out in (He et al., 2021), the expression power of GNNs is highly associated with the choice of polynomials that are used to approximate the graph filter. In this paper, from the perspective of linear algebra, we claim that any polynomial of order $K$ has the same expression ability, which can be mathematically explained by the following theorem.

**Theorem 3.1** Let $g_1(x), g_2(x), \ldots, g_n(x)$ be $n$ polynomials of the form $\sum_{i=0}^{k} \alpha_i x^i$. The maximum linearly independent group of these $n$ polynomials has $K + 1$ elements, and then it can span the linear polynomial space of degree $K$.

The proof of Theorem 3.1 can be found in the Appendix A.1. This theorem implies that any $K$ degree polynomial filter can be exactly represented by a set of $K$ order linearly independent polynomial bases. Since GNNs consist of successive layers as shown in Eq. (1), GNNs can thus be viewed as an optimizer to solve a convex optimization problem as shown in Eq. (4) with a globally optimal solution (Please refer to the proof of Theorem 3.1 in the Appendix A.1.). Specifically, Theorem 3.1 indicates that no matter whether a $K$-order polynomial is used as the basis (i.e. $\boldsymbol{C}^{(s)}$), it can converge to a global optimal point. That is to say, although different polynomial bases are used to approximate the graph signal filter, their expressive power will be the same if all polynomials have the same degree.

### 3.1.2 CONVERGENCE RATE ANALYSIS OF GNN

After discussing the expressive ability of GNNs, a natural question is: how quickly can GNNs converge to the optimal solution? Regarding convergence rate, the mainstream idea is that selecting an orthogonal basis will affect the condition number of Hessian matrix and thus affect the learning effect and convergence speed (Guo & Wei, 2023). In this work, we make a preliminary attempt to study this effect from the perspective of solving one convex optimization using the gradient descent (GD) algorithm. Specifically, we first prove that this effect does not exceed a linear level and then provide an upper bound of the convergence rate. The analysis process is shown below.

Based on previous works, GNNs can be seen as an optimizer and can be mathematically formulated in a uniform optimization form ((Dengyong Zhou, 2003; Zhu et al., 2021; He et al., 2021))):

$$\min_{\boldsymbol{z}} f(\boldsymbol{z}) = (1 - \alpha)\boldsymbol{z}^T \gamma(\boldsymbol{L})\boldsymbol{z} + \alpha \|\boldsymbol{z} - \boldsymbol{x}\|_2^2, \tag{4}$$

where $\boldsymbol{z}$ is the output feature matrix, $\gamma(\boldsymbol{L})$ is an energy function of the positive-definite graph Laplacian matrix $\mathbf{L}$, $\boldsymbol{x}$ is the label matrix, and $\alpha \in [0, 1)$. Generally, $\gamma(\cdot)$ operates on the spectral of $\boldsymbol{L}$, and we have $\gamma(L) = \boldsymbol{U} diag[\gamma(\lambda_1), \ldots, \gamma(\lambda_n)]\boldsymbol{U}^\top$. Fortunately, Eq. (4) is a convex optimization problem, and thus its global optimal solution exists. Moreover, since this optimization problem is quadratic, its convergence rate can be numerically analyzed using the following inequality.

**Lemma 3.1 (Kantorovich inequality) ((Marshall & Olkin, 1990; Liu & Neudecker, 1999; Liu et al., 2022))** For any $\boldsymbol{z} \in \mathbb{R}^n$, denote $\lambda_{min}$ and $\lambda_{max}$ as the maximum and minimum eigenvalues of a positive definite symmetric matrix $\boldsymbol{M} \in \mathbb{R}^{n \times n}$, respectively, and then

$$\frac{(\boldsymbol{z}^T \boldsymbol{M} \boldsymbol{z})(\boldsymbol{z}^T \boldsymbol{M}^{-1} \boldsymbol{z})}{(\boldsymbol{z}^T \boldsymbol{z})^2} \leq \frac{(\lambda_{\max} + \lambda_{\min})^2}{4\lambda_{\max}\lambda_{\min}}. \tag{5}$$

Using Lemma 3.1, we can prove the following convergence rate theorem for GNNs.

**Theorem 3.2 (Convergence rate theorem for GNN)** The convergence rate of the gradient descent algorithm with exact line search to solve the optimization problem (4) is not faster than a linear constant. Mathematically, we have the following convergence formula:

$$\left| \frac{f(\boldsymbol{z}_{k+1}) - f(\boldsymbol{z}^*)}{f(\boldsymbol{z}_k) - f(\boldsymbol{z}^*)} \right| \leq (1 - \frac{2}{1 + \kappa})^2, \tag{6}$$

where $\boldsymbol{z}_k$ is the update of the $k^{th}$ iteration, $\boldsymbol{z}^*$ is the global minimum solution, and $\kappa$ is the condition number of $2((1-\alpha)\gamma(\boldsymbol{L}) + \alpha\boldsymbol{I})$.

Theorem 3.2 implies that the convergence rate difference of the optimization problem (4) underlying GNNs is no more than a linear constant related to the condition number of the problem matrix. Essentially, the impact of setting different polynomial bases is reflected in $\kappa$. As can be seen, the minimum of $\kappa$ is 1. When $\kappa = 1$ and $\alpha = 1$, Eq. (4) degrades into a trivial regression problem, i.e. $\min\limits_{\boldsymbol{z}} f(\boldsymbol{z}) = \|\boldsymbol{z} - \boldsymbol{x}\|_2^2$, whose optimal solution is $\boldsymbol{z} = \boldsymbol{x}$. That degradation case also implies that the optimizing of condition number also has a risk of worsening the structure of the problem. The proof of Theorem 3.2 can be found in the Appendix A.1.

Moreover, since selecting the optimal orthogonal basis requires an additional expensive calculation cost (Guo & Wei, 2023), we therefore make an elementary attempt to replace these complex orthogonal polynomial bases with a simpler monomial basis. Specifically, monomial bases, i.e. $(2\boldsymbol{I} - \boldsymbol{L})^k, k = 1, 2, ..., K$, are designed to approximate the graph filter and we surprisingly find that when the weights are all fixed as 1 (we call the GNN FixedMono), FixedMono outperforms BernNet on homogeneous graph datasets with less computational cost as shown in Table 1. This inspires us to build a more interpretable basis since existing GNNs mainly focus on searching for some highly restricted polynomial bases, while largely ignoring that designing polynomial or monomial bases to approximate the targeted graph filters should also incorporate graph structure information in order to better utilize the graph information.

Table 1: Comparison between Bernstein basis and FixedMono basis on homogeneous datasets.

| Dataset | Cora | Computers | Pubmed | citepseer | Photo |
|---|---|---|---|---|---|
| BernNet | $88.52 \pm 0.95$ | $87.64 \pm 0.44$ | $88.48 \pm 0.41$ | $80.09 \pm 0.79$ | $93.63 \pm 0.35$ |
| FixedMono (Ours) | $\mathbf{88.76 \pm 0.90}$ | $\mathbf{89.15 \pm 0.42}$ | $\mathbf{88.84 \pm 0.48}$ | $\mathbf{80.34 \pm 0.70}$ | $\mathbf{95.08 \pm 0.35}$ |

## 3.2 SOLUTIONS TO PROBLEM 2

For the second problem, we first propose a Graph Information Activation (GIA) theory, which can be used to evaluate various popular bases. Then, we design a new basis embedded with the graph structure information and this basis can be theoretically explained by the GIA theory. Based on this basis, we further propose a simple GNN (i.e. SimpleNet).

### 3.2.1 GRAPH INFORMATION ACTIVATION THEORY

Inspired by graph theory (Andrilli & Hecker, 2023; Bronson et al., 2024), polynomial filter and message passing theory on the graph (Gilmer et al., 2017; Yi Liu, 2022), we propose the Graph Information Activation (GIA) theory in this section. Before introducing the main theorem, we first introduce some necessary definitions and lemmas briefly.

**Definition 3.1 ($K$-step neighborhood)** For an undirected and unweighted graph, the $K$-step neighborhood of node $v$ is a set consisting of nodes which have a $K$-length simple path to the node $v$. The $K$-step neighborhood of node $v$ is denoted as $N_k(v)$.

**Definition 3.2 ($K$-step activation)** For an undirected and unweighted graph, the $K$-step activation $\boldsymbol{x}_v^*$ of a given node $v$ equals the linear combination of the feature vector $\boldsymbol{x}_0$ of node $v$ and all its one-step to K-step neighbours' feature vectors. Mathematically, the $K$-step activation is defined as

$$\boldsymbol{x}_v^* = \sum_{k=1}^{K} \sum_{s \in N_k(v)} \alpha_s \boldsymbol{x}_s + \alpha_0 \boldsymbol{x}_0 \tag{7}$$

where $\boldsymbol{x}_v^*$ is the $K$-step activation of node $v$, $\alpha_s$ and $\alpha_0 (\geq 0)$ are combination coefficients, and $\boldsymbol{x}_s$ is the feature vector of node $s$.

**Definition 3.3 (Positive activation)** In Eq. (7), if all the $\alpha_s s$ are non-negative and there is at least one greater than 0, then the $K$-step activation is called positive; otherwise, it is non-positive.

**Definition 3.4 (Proper activation)** In Eq. (7), if $\alpha_0 > 0$, then the $K$-step activation is proper; otherwise, it's improper.

Here, we give an explanation for the two properties. Specifically, the positive property describes the positive feedback between neighbours in a graph, and the proper property means that one vertex updated after activation contains its own feature. It can be viewed as a residual connection in the activation of a graph, which inclines to make the activation more stable and consistent. The activation illustrations are shown in Figure 1.

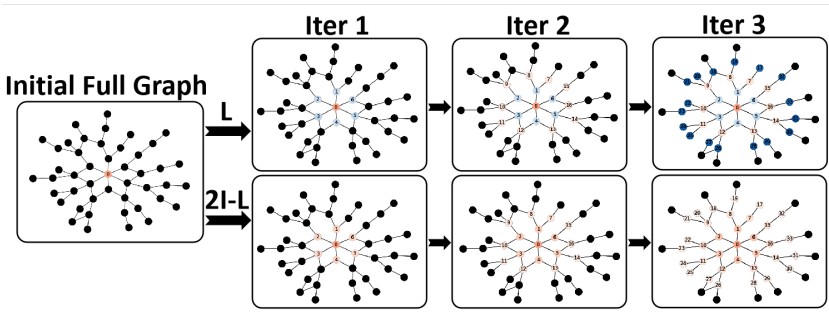

Figure 1: **Left:** Schematic depiction of a given full graph. The central node is the observing node with an initial weight of one. We focus on its two activations. **Right:** The black nodes are those that haven't been activated. The red nodes are those with positive weights. The blue nodes are those with negative weights. The redder or bluer the node, the greater the absolute value of its weights. We show two activations $\boldsymbol{L}$ (upper) and $(2\boldsymbol{I} - \boldsymbol{L})$ (lower) with 3 iterations, primary, quadratic and cubic terms. In this figure, $(2\boldsymbol{I} - \boldsymbol{L})^k$ is a positive and proper activation while $\boldsymbol{L}$ is non-proper.

**Definition 3.5 (Graph activation)** For a graph $G(V, E)$ with $|V| = n$ and feature matrix $\boldsymbol{X} \in \mathbb{R}^{n \times d}$, where $d$ is the dimension of node feature, the activation of $G$ can be computed as $\boldsymbol{X}^* = \boldsymbol{A}\boldsymbol{X}$, where $\boldsymbol{A}$ is a transformation matrix and $\boldsymbol{X}_{i:}^*$ is the activation of the $i^{th}$ node in the graph.

As defined in Eq. (7), activation of a node can be seen as the linear reconstruction of its neighbours and itself, and activation of a whole graph can be seen as a special linear transformation of features. In the rest of this section, we mainly focus on the polynomial forms of this transformation; that is to say, we are dedicated to constructing an activation in polynomial form which is capable of finely extracting graph information as previous GNNs do. The novelty is that this time, we are not trying to approximate some spectral filter in polynomials. Conversely, we attempt to construct a polynomial spatial filter from a GIA perspective.

Generally, the graph can be divided into two categories, i.e. homogeneous graph and nonhomogeneous graph. Let's start with the simpler case, i.e., homogeneous graph. Due to the positive correlation of neighbouring nodes in homogeneous graphs, a proper and positive activation can be efficient for feature extraction. Now consider a problem: Can we find a proper and positive activation in polynomial form for all nodes in a graph? Let us refer to the following lemmas and theorems to answer this question.

**Lemma 3.2.** $G(V, E)$ is a graph with self-loop, $|V| = n$, and $X$ is the feature matrix of $G$. Let $\boldsymbol{A}$ be a $n \times n$ matrix such that

$$\begin{cases} \boldsymbol{A}_{ij} > 0 & \text{if } (v_i, v_j) \in E, \\ \boldsymbol{A}_{ij} = 0 & \text{else} \end{cases}$$

and then $\boldsymbol{X}^* = \boldsymbol{A}^k \boldsymbol{X}$ is a positive and proper $k$-step activation of graph $G$.

**Lemma 3.3.** The sum of finite positive and proper activations is still positive and proper.

Based on Lemma 3.2 and Lemma 3.3, we can then obtain the following theorem.

**Theorem 3.3.** Let $\boldsymbol{L}$ be the Laplacian matrix of graph $G$ and $\boldsymbol{I}$ be the identity matrix, and then $\sum_{j=0}^{k} \alpha_j (2\boldsymbol{I} - \boldsymbol{L})^j X, (\forall \alpha_j \geq 0)$, is a proper and positive activation of $G$.

We now get a general form of positive and proper activation of a graph as we expect initially. Also, our FixedMono model well fits this form. Next, we will present the GIA analysis on the FixedMono base and other popular bases to better understand the GIA theory and how it works. All the proof can be found in Appendix A.1.

### 3.2.2 ANALYSIS ON FIXEDMONO AND OTHER POPULAR BASES USING GIA THEORY

Let's look back on the FixedMono basis used in Section 3.1.2, which performs well in homogeneous graph datasets as shown in Table 1 since the transform matrix constructed by the FixedMono base can generate a positive and proper activation of a graph. The positive nature of the FixedMono makes it slightly outperform many main baselines, as the neighbourhood of one vertex is too large as $k$ increases, making some neighbours not see it as 'nearby'. This problem is also called the "curse of dimensionality" (Bellman & Corporation, 1957; Bellman, 1961) in the field of machine learning.

However, it should also be noted that FixedMono does not work well on non-homogeneous graphs, as shown in Table 3, which may be attributed to the lack of negative feedback in activation. As we know, the non-homogeneous graph consists of heterogeneous components. Two neighbours in different classes may have negative feedback, that is to say, in some cases if two vertexes have a connected edge, they have a high probability of being classified into different classes, e.g., if one paper cites another one, they must be published at time-varying points.

Furthermore, we can generally conduct analysis on the popular polynomial bases using GIA theory. For example, we focus on analyzing the Monomial $(1 - \lambda)^k$ and Bernstein $(2 - \lambda)^{K-k}\lambda^k$ bases. Using the GIA theory, we can find that the $(\boldsymbol{I} - \boldsymbol{L})^k$ base contains pure positive information activation but lacks a non-positive component, thus limiting its expressive power. Also, $(\boldsymbol{I} - \boldsymbol{L})^1$ is improper due to the lack of self-loop. $(2\boldsymbol{I} - \boldsymbol{L})^{K-k}\boldsymbol{L}^k$ base really contains positive and non-positive information activation. However, this mixture of positive and non-positive components does not decouple the learning process of positivity and non-positivity, that is to say, we cannot optimize the positive or non-positive activation independently in our network. The only learned leading coefficient in the term $(2\boldsymbol{I} - \boldsymbol{L})^{K-k}\boldsymbol{L}^k$ will scale up or hold both positive and non-positive components simultaneously, which also forces us to find a simpler way to decouple the positive and non-positive components in our activation.

### 3.2.3 SIMPLENET BASED ON GIA THEORY

Inspired by the GIA theory and FixedMono's failure on the non-homogeneous datasets, we propose a new GNN called SimpleNet. To understand the SimpleNet better, we first perceive it from the classic spectral-GNN perspective. Given an arbitrary filter function $h : [0, 2] \rightarrow [0, 1]$, the spectral filter on the graph signal $x$ is denoted as

$$h(\boldsymbol{L})\boldsymbol{x} = \boldsymbol{U}h(\Lambda)\boldsymbol{U}^T\boldsymbol{x} = \boldsymbol{U}diag[h(\lambda_1), ..., h(\lambda_n)]\boldsymbol{U}^T\boldsymbol{x}. \tag{8}$$

In particular, given a graph signal $\boldsymbol{x}$, the convolution operator of our SimpleNet in the Laplacian spectrum is defined as

$$\boldsymbol{z} = (\sum_{i=0}^{K_1} \alpha_i(2\boldsymbol{I} - \boldsymbol{L})^i + \sum_{j=0}^{K_2} \beta_j \boldsymbol{L}^j)\boldsymbol{x}, \tag{9}$$

where $K_1$ and $K_2$ are two hyper-parameters, $\alpha_i$ and $\beta_j$ are the weight coefficients of the basis. Furthermore, the proposed basis used in Eq. (9) can be explained by our GIA theory. Specifically, $(2\boldsymbol{I} - \boldsymbol{L})^i$ is a positive and proper activation of graph information. To make it adaptive to non-homogeneous datasets, we need another injection of non-positive components; thus, we simply add the $\boldsymbol{L}^j$ term into our basis, which is a non-positive activation. In summary, our SimpleNet uses the weighted mixture of positive and non-positive information activations simultaneously to achieve proper information fusion, and the weight parameters (i.e., $\alpha_i$ and $\beta_j$) are learnable. Furthermore, the positive and non-positive ratios can be rectified by adjusting $K_1$ and $K_2$. Also, note that when SimpleNet performs convolution operations on the graph, it is equivalent to performing appropriate information activation, and this process is shown in Figure 1.

## 4 EXPERIMENTAL RESULTS

In this section, we conduct experiments on real-world datasets to evaluate the performance of our proposed SimpleNet. All experiments are carried out on a machine with an RTX 3090 GPU (24GB memory), Intel Xeon CPU (2.20 GHz), and 256GB of RAM.

Table 2: The statistics of our used real datasets for node classification.

| Datasets | Cora | Computers | Photo | Texas | Cornell | Citeseer | Actor | Chameleon | Pubmed | Penn94 |
|---|---|---|---|---|---|---|---|---|---|---|
| Nodes | 2708 | 13752 | 7650 | 183 | 183 | 3327 | 7600 | 2277 | 19717 | 41554 |
| Edges | 5278 | 245861 | 119081 | 279 | 277 | 4552 | 26659 | 31371 | 44324 | 1362229 |
| Features | 1433 | 767 | 745 | 1703 | 1703 | 3703 | 932 | 2325 | 500 | 5 |
| Classes | 7 | 10 | 8 | 5 | 5 | 6 | 5 | 5 | 5 | 2 |

Table 3: Results on homogenous datasets: mean accuracy (%) ± 95% confidence interval.

| | Computers | Photo | Pubmed | Citeseer | Cora |
|---|---|---|---|---|---|
| GCN | 83.32±0.33 | 88.26±0.73 | 86.74±0.27 | 79.86±0.67 | 87.14±1.01 |
| ChebNet | 87.54±0.43 | 93.77±0.32 | 87.95±0.28 | 79.11±0.75 | 86.67±0.82 |
| GPRGNN | 86.85±0.25 | 93.85±0.28 | 88.46±0.33 | 80.12±0.83 | 88.57±0.69 |
| BernNet | 87.64±0.44 | 93.63±0.35 | 88.48±0.41 | 80.09±0.79 | 88.52±0.95 |
| JocobiConv | 90.39±0.29 | 95.43±0.23 | 89.62±0.41 | 80.78±0.79 | 88.98±0.46 |
| ChebNetII | 89.37±0.38 | 94.53±0.25 | 88.93±0.29 | 80.53±0.79 | 88.71±0.93 |
| OptBasis | 89.65±0.25 | 93.12±0.43 | 90.30±0.19 | 80.58±0.82 | 88.02±0.70 |
| FixedMono (ours) | 89.15±0.42 | 95.08±0.35 | 88.84±0.48 | 80.34±0.70 | 88.76±0.90 |
| LearnedMono (ours) | 90.37±0.32 | 95.44±0.25 | 89.38±0.52 | 80.39±0.69 | 88.85±0.80 |
| SimpleNet (ours) | **90.42±0.30** | **95.46±0.31** | **91.06±0.34** | **80.84±0.56** | **89.01±0.38** |

## 4.1 NODE CLASSIFICATION ON REAL-WORLD DATASETS

**Datasets.** We evaluate the performance of our SimpleNet on real-world datasets. Following (He et al., 2021) and (Guo & Wei, 2023), we adopt 5 homogeneous graphs, i.e. Cora, Citeseer, Pubmed, Computers and Photo, and 5 non-homogeneous graphs, i.e. Chameleon, Actor, Texas, Cornell and Penn94 in our experiments. The statistics of these datasets are summarized in Table 2.

**Experimental Setup.** We perform a full-supervised node classification task, where we randomly split each data set (except Penn94) into a train/validation/test set with a ratio of 60%/20%/20%. For Penn94, we use the partitioned data sets given in (Guo & Wei, 2023). For Chameleon, Actor, and Penn94, the experiment setup is the same as (Guo & Wei, 2023). For the other datasets, the experiment setup is the same as (He et al., 2021). More detailed settings can be found in Appendix A.2.

In all the experiments, our proposed methods contain three variations, i.e., FixedMono, Learned-Mono, and SimpleNet, which are defined as follows. FixedMono uses a monomial basis ($(2\boldsymbol{I} - \boldsymbol{L})^k, k = 1, 2, \ldots, K$) with positive activation property and weights are fixed with 1. LearnedMono makes the weights of FixedMono learnable. SimpleNet adds the non-positive activation term into the LearnedMono as Eq. (9). Besides, the competing methods are GCN (Kipf & Welling, 2017), ChebNet (Defferrard et al., 2016), GPR-GNN (Chien et al., 2021), BernNet (He et al., 2021), JacobiConvs (Wang & Zhang, 2022), ChebNetII (He et al., 2022), OptBasisGNN (Guo & Wei, 2023). The micro-F1 score with a 95% confidence interval is used as the evaluation metric. The polynomial degree $K$ in other competing methods is set following their original papers. For our SimpleNet, the degree parameters (i.e., $K_1$ and $K_2$) are set between 0 and 6.

**Results.** The experimental results on homogeneous graphs and non-homogeneous graphs are summarized in Table 3 and Table 4, respectively. As can be seen from Table 3, our SimpleNet can consistently obtain the best performance on all five homogenous datasets. Additionally, by comparing FixedMono and LearnedMono, we can observe that LearnedMono can significantly improve the performance of FixedMono, which can be attributed to the learnable weights that make the model more flexible to adapt to the dataset. Furthermore, by comparing LearnedMono and SimpleNet, it can be seen that SimpleNet can further promote the performance of LearnedMono. This should be rationally explained by the injected non-positive activation in the SimpleNet that makes the model better adapt to some real pattern in the homogenous graph. Table 4 records the performance comparison of all the methods on non-homogeneous datasets. Specifically, our SimpleNet can achieve the best results on Texas and Cornell datasets and comparable results on Actor and Penn94 datasets compared with other SOTA methods. It should also be noted that the performance of our SimpleNet on the Chameleon dataset is worse than that of some competing methods, which may be due to the high-dimensional node feature in this dataset. Further, by comparing LearnedMono and SimpleNet, we can see that SimpleNet can significantly boost the performance of LearnedMono, which verifies that non-positive activation is really beneficial for the non-homogeneous graphs.

Table 4: Results on non-homogenous datasets: mean accuracy (%) ± 95% confidence interval.

| | Texas | Cornell | Actor | Chameleon | Penn94 |
|---|---|---|---|---|---|
| GCN | 77.38±3.28 | 65.90±4.43 | 33.26±1.15 | 60.81±2.95 | 82.47±0.27 |
| ChebNet | 86.22±2.45 | 83.93±2.13 | 37.42±0.58 | 59.51±1.25 | 82.59±0.31 |
| GPRGNN | 92.95±1.31 | 91.37±1.81 | 39.91±0.62 | 67.49±1.37 | 83.54±0.32 |
| BernNet | 93.12±0.65 | 92.13±1.64 | 41.71±1.12 | 68.53±1.68 | 83.26±0.29 |
| JocobiConv | 93.44±2.13 | 92.95±2.46 | 41.17±0.64 | 74.20±1.03 | / |
| ChebNetII | 93.28±1.47 | 92.30±1.48 | 41.75±1.07 | 71.37±1.01 | **84.86±0.33** |
| OptBasis | 90.62±3.44 | 88.77±4.75 | **42.39±0.52** | **74.26±0.74** | 84.85±0.39 |
| FixedMono (ours) | 77.70±3.12 | 74.59±3.20 | 40.97±1.90 | 55.23±1.70 | 82.03±0.35 |
| LearnedMono (ours) | 74.75±7.38 | 79.18±2.29 | 41.17±1.25 | 65.73±2.18 | 79.67±0.37 |
| SimpleNet (ours) | **95.73±0.98** | **94.19±1.31** | 40.97±1.90 | 68.09±1.64 | 82.64±0.43 |

Table 5: Average running time per epoch (ms)/average total running time (s).

| | Cora | Citeseer | Pubmed | Computer | Photo | Texas | Cornell |
|---|---|---|---|---|---|---|---|
| GCN | 5.59/1.62 | 4.63/1.95 | **5.12/1.87** | **5.72/2.52** | **5.08/2.63** | **4.58/0.92** | 4.83/0.97 |
| APPNP | 7.16/2.32 | 7.79/2.77 | 8.21/2.63 | 9.19/3.48 | 8.69/4.18 | 7.83/1.63 | 8.23/1.68 |
| ChebNet | 6.25/1.76 | 8.28/2.56 | 18.04/3.03 | 20.64/9.64 | 13.25/7.02 | 6.51/1.34 | 5.85/1.22 |
| GPRGNN | 9.94/2.21 | 11.16/2.37 | 10.45/2.81 | 16.05/4.38 | 13.96/3.94 | 10.45/2.16 | 9.86/2.05 |
| BernNet | 19.71/5.47 | 22.36/6.32 | 22.02/8.19 | 28.83/8.69 | 24.69/7.37 | 23.35/4.81 | 22.23/5.26 |
| JocobiConv | 6.40/3.10 | 6.30/3.00 | 6.60/4.90 | 7.30/4.80 | 6.40/4.80 | 6.60/3.40 | 6.50/3.40 |
| OptBasis | 19.38/7.57 | 13.88/4.53 | 39.60/15.79 | 16.16/13.49 | 13.25/10.59 | 14.07/8.61 | 13.03/8.16 |
| FixedMono (ours) | **4.59/1.17** | **4.95/0.78** | 3.95/2.13 | 5.15/3.30 | 5.31/3.84 | 5.33/2.68 | 5.35/2.69 |
| LearnedMono (Ours) | 6.44/1.64 | 5.96/0.94 | 4.51/2.89 | 5.52/3.96 | 5.58/3.97 | 7.07/3.55 | 7.50/3.77 |
| SimpleNet (ours) | 7.15/8.61 | 7.00/2.46 | 5.02/2.53 | 6.00/3.46 | 6.15/5.26 | 6.12/1.28 | 3.81/1.54 |

The training time for each method is shown in Table 5. As can be seen, our three proposed methods have a lower average running time and a lower total running time than most other methods, and thus they should be more efficient and user-friendly in practice. This may be attributed to the following reasons. Firstly, since the multiplication operation is closed, i.e. $(2\boldsymbol{I}-\boldsymbol{L})^{k-1}*(2\boldsymbol{I}-\boldsymbol{L}) = (2\boldsymbol{I}-\boldsymbol{L})^{k}$, the calculation of the monomial basis is thus simpler than general polynomials. Specifically, the computational time complexity is $O(n)$. Secondly, since our basis does not require a big degree parameter, the parameter learning of our methods is relatively easier than other competing methods. More comparisons on running time can be found in Appendix A.3.

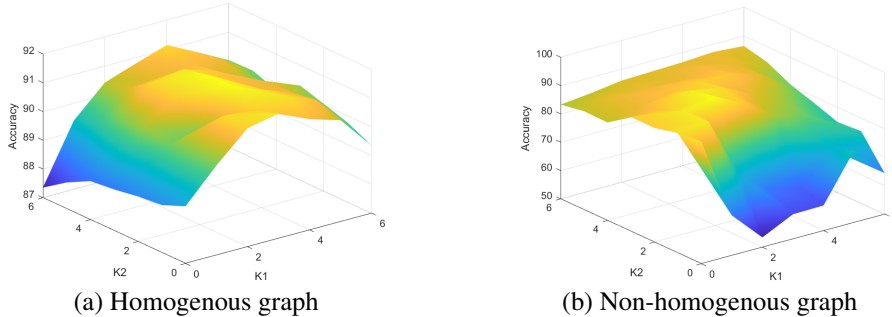

|  (a) Homogenous graph  |  (b) Non-homogenous graph  |

Figure 2: Accuracy of different degree parameters ($K_1$ and $K_2$) for our SimpleNet.

## 4.2 MORE DISCUSSIONS

**Effect of learnable weight (i.e. fixed weight vs. learnable weight).** The necessity of setting the weight learnable can be explained by comparing the experimental results of FixedMono and LearnedMono presented in Table 3 and Table 4. As can be seen, if the weights are learnable, the performance can be further improved compared with the fixed weight.

**Effect of setting different degree parameters ($K_1$ and $K_2$).** The results are shown in Figure 2. For the homogeneous graph as shown in Figure 2(a), when $K_1$ is fixed and $K_2$ varies, the accuracy does not change so rapidly. When $K_2$ is fixed and $K_1$ varies, the accuracy changes relatively more rapidly. Generally, our SimpleNet is not sensitive to the degree parameters on homogeneous graphs. For the non-homogeneous graph as shown in Figure 2(b), we can observe that our SimpleNet is

more sensitive to the parameters. No matter whether $K_1$ or $K_2$ is fixed, the accuracy will be affected obviously. The higher accuracy can be obtained when $K_1$ and $K_2$ are close.

**Effect of $K$-step activation's properties.** To verify the effect of K-step activation's property, e.g. positive property, we further conduct a comparative experiment on the three bases, i.e., $\sum_{i=0}^{K}(2\boldsymbol{I}-\boldsymbol{L})^i$, $\sum_{i=0}^{K}\boldsymbol{L}^i$ and $\sum_{i=0}^{K_1}(2\boldsymbol{I}-\boldsymbol{L})^i + \sum_{j=0}^{K_2}\boldsymbol{L}^j$. The experimental results are shown in Figure 3. As can be seen, due to its positive nature, the basis $2\boldsymbol{I}-\boldsymbol{L}$ is more powerful in tackling the homogeneous graph while less effective in dealing with the non-homogeneous graph compared to the non-positive basis $\boldsymbol{L}$. Moreover, from Figure 3, we can observe that our SimpleNet combining the two bases together evidently outperforms the two pure bases.

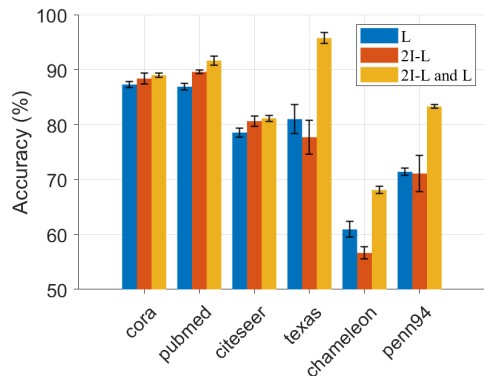

Figure 3: Performance comparison of different activations.

## 5 LIMITATIONS AND FUTURE WORK

**Weakness on large scale non-homogeneous datasets.** As shown in Table 3, our SimpleNet cannot always perform best in large-scale non-homogeneous datasets, such as Penn94 and Chameleon. This may be caused by these reasons. 1) As feature dimension and edge number increase, the "curse of dimensionality" starts to have a big impact. 2) In large-scale non-homogeneous datasets, connectivity and locality are harder to characterize. These shortcomings require discovering powerful activation with more properties.

**More in-depth discussion on GIA theory.** Except for positive and proper properties, more properties of activations should be explored, e.g., non-positive properties need a more detailed introduction. Besides, only the linear case of graph information activation is considered in this study. However, real graph information activation may be non-linear, thus the non-linear case should be considered.

**Limiting assumption.** All the neighbours are uniformly processed in a graph, which implies a hidden assumption that all the same-step neighbours are equally treated in the feature space and the activation is thus only concerned with the graph structure. However, this assumption may cause a loss of generality since the feature itself should contribute to the intensity of activation. In the future, we will attempt to relax this assumption and allow the network to more flexibly learn different weights to the same-step neighbours.

## 6 CONCLUSION

In this paper, we have first made a theoretical analysis on the expressive power and convergence rate of GNNs and found that finely setting a polynomial basis by only considering the polynomial property can bring limited benefits. Then, we proposed a graph information activation (GIA) theory that provides a new perspective for the interpretation and design of polynomial filters in GNN and analysed some popular bases using the GIA theory. Based on the GIA theory and analysis, we design a new basis by considering the graph information and further build a new GNN (i.e., SimpleNet), which can be applied to both homogeneous and non-homogenous graphs. Experimental results illustrate that our SimpleNet performs better and faster than most other popular GNNs.

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

# A   APPENDIX

## A.1   PROOF OF THEOREMS

**Theorem 3.1** Let $g_1(x), g_2(x), \ldots, g_n(x)$ be n polynomials of the form $\sum_{i=0}^{k} \alpha_i x^i$. The maximum linearly independent group of these $n$ polynomials has $K + 1$ elements, then it can span the polynomial linear space of degree $K$.

***Proof:*** when $K = 1$, $g_o = \alpha_0 (\alpha \neq 0)$, now obviously that the constant $\alpha_0$ can span the polynomial space of degree 0.

Considering the case $K \geq 1$. Notice that each element a in k degree polynomial space can be written as the form

$$a = \alpha_0 + \alpha_1 x + ... + \alpha_k x^k \tag{10}$$

so in this basis, a can be represented uniquely as a $k+1$-dimension vector as below

$$a = (\alpha_0, \alpha_1, ..., \alpha_k)^T \tag{11}$$

Actually, the Eq. (11) can be written as a bijection mapping $k$ degree polynomial space to $k+1$-dimension linear space, thus we can say that the two spaces are isomorphic to each other. Then our theorem is equivalent to the statement: Let $y_1, y_2, ..., y_n$ be $n$ elements of the $k+1$-dimension linear space. The maximum linearly independent group of these n polynomials has $K+1$ elements, then it can span the $k+1$-dimension linear space.

From linear algebra, we can see that the statement above is obviously true, as all $k+1$ independent $k+1$-dimension vectors can span $k+1$ dimension linear space.

**Lemma 3.1 (Kantorovich inequality) ((Marshall & Olkin, 1990; Liu & Neudecker, 1999; Liu et al., 2022))** For any $z \in \mathbb{R}^n$, denote $\lambda_{min}$ and $\lambda_{max}$ as the maximum and minimum eigenvalues of a positive definite symmetric matrix $M \in \mathbb{R}^{n \times n}$, respectively, then

$$\frac{(z^T M z)(z^T M^{-1} z)}{(z^T z)^2} \leq \frac{(\lambda_{\max} + \lambda_{\min})^2}{4\lambda_{\max}\lambda_{\min}}. \tag{12}$$

***Proof:*** Considering that $G$ is positive definite and symmetric, we can conduct eigenvalue decomposition of G as follows:

$$G = U^T \Lambda U, \tag{13}$$

where $U$ is a $n \times n$ orthogonal matrix. Let $y = Uz$, $a_i = y_i^2$, then we can rewrite the left term in inequality 13

$$\frac{(z^T G z)(z^T G^{-1} z)}{(z^T z)^2} = \frac{(z^T U^T \Lambda U z)(z^T U^T \Lambda^{-1} U z)}{(z^T U^T U z)^2} \tag{14}$$

$$= \frac{(y^T \Lambda y)(y^T \Lambda^{-1} y)}{(y^T y)^2} \tag{15}$$

$$= \frac{(\sum_{i=1}^{n} \lambda_i y_i^2)(\sum_{i=1}^{n} \lambda_i^{-1} y_i^2)}{(\sum_{i=1}^{n} a_i^2)^2}. \tag{16}$$

Without loss of generality, we let $\sum_{i=1}^{n} a_i = 1$, then the inequality we want to prove is equivalent to

$$(z^T G z)(z^T G^{-1} z) \leq \frac{(\lambda_{\max} + \lambda_{\min})^2}{4\lambda_{\max}\lambda_{\min}},$$

where the $\lambda$ is notated in ascending order, which means $\lambda_1 =_{min}$, and $\lambda_n = \lambda_{max}$. Now we can construct a quadratic function

$$f(x) = (\sum_{i=1}^{n} \lambda_i^{-1} a_i) x^2 - \frac{\lambda_1 + \lambda_n}{\sqrt{\lambda_1 \lambda_n}} x + (\sum_{i=1}^{n} \lambda_i a_i). \tag{17}$$

We can see that the inequality is true if and only if the discriminant of f(x) $\Delta \geq 0$, which is to say that f(x) has zero over the field of real numbers. Then, we construct

$$f(\sqrt{\lambda_1 \lambda_n}) = \lambda_1 \lambda_n \sum_{i=1}^{n} \lambda_i^{-1} a_i - (\lambda_1 + \lambda_n) + \sum_{i=1}^{n} \lambda_i a_i \tag{18}$$

$$= \lambda_1 \lambda_n (\lambda_1^{-1} a_1 + \lambda_n^{-1} a_n + \sum_{i=2}^{n-1} \lambda_i a_i) - (\lambda_1 + \lambda_n) + \lambda_1 a_1 + \lambda_n a_n + \sum_{i=2}^{n-1} \lambda_i a_i \tag{19}$$

$$= (a_1 + a_n - 1)(\lambda_1 + \lambda_n) + \lambda_1 \lambda_n \sum_{i=2}^{n-1} \lambda_i^{-1} a_i + \sum_{i=2}^{n-1} \lambda_i a_i \tag{20}$$

$$= (a_1 + a_n - \sum_{i=1}^{n} a_i)(\lambda_1 + \lambda_n) + \lambda_1 \lambda_n \sum_{i=2}^{n-1} \lambda_i^{-1} a_i + \sum_{i=2}^{n-1} \lambda_i a_i \tag{21}$$

$$= -(\lambda_1 + \lambda_n) \sum_{i=2}^{n-1} a_i + \lambda_1 \lambda_n \sum_{i=2}^{n-1} \lambda_i^{-1} a_i + \sum_{i=2}^{n-1} \lambda_i a_i \tag{22}$$

$$= \sum_{i=2}^{n-1} \left\{ a_i (\lambda_1 \lambda_n \lambda_i^{-1} + \lambda_i - \lambda_1 - \lambda_n) \right\} \tag{23}$$

$$= \sum_{i=2}^{n-1} \left\{ a_i \lambda_i^{-1} (\lambda_1 \lambda_n + \lambda_i^2 - (\lambda_1 + \lambda_n)\lambda_i) \right\} \tag{24}$$

$$= \sum_{i=2}^{n-1} \left\{ a_i \lambda_i^{-1} (\lambda_i - \lambda_1)(\lambda_i - \lambda_n) \right\} \tag{25}$$

$\lambda_1 \leq \lambda_i \leq \lambda_n$, so $(\lambda_i - \lambda_1)(\lambda_i - \lambda_n) \leq 0$. $G$ is positive definite, so $\lambda_i \geq 0$. $a_i = y_i^2 \geq 0$. So $\sum_{i=2}^{n-1} \left\{ a_i \lambda_i^{-1} (\lambda_i - \lambda_1)(\lambda_i - \lambda_n) \right\} \leq 0$ and $f(\sqrt{\lambda_1 \lambda_n}) \leq 0$. Thus, the quadratic function f(x) has zero in the real number field and the characteristic of f(x), $\geq 0$. Finally, the inequality is true. Our proof ends.

**Theorem 3.2 (Convergence rate theorem for GNNs)** The convergence rate of gradient descent algorithm with exact line search to solve the optimization problem (4) is no faster than a linear constant. Mathematically, we have the following convergence formula:

$$\left| \frac{f(z_{k+1}) - f(z^*)}{f(z_k) - f(z^*)} \right| \leq (1 - \frac{2}{1 + \kappa})^2, \tag{26}$$

where $z_k$ is the update of the $k_{th}$ iteration, $z^*$ is the global minimum solution, and $\kappa$ is the condition number of $2((1 - \alpha)\gamma(L) + \alpha I)$.

***Proof:*** The problem (4) is a convex optimization problem and has a globally optimal solution. We denote the global optimal solution as $z^*$ and denote $\frac{1}{2}Q = (1 - \alpha)\gamma(L) + \alpha I$, $b^T = -2\alpha x^T$. We define the convergence rate of the problem below and we can write the problem as:

$$z^* = \arg\min_z h(z) = \frac{1}{2}z^T Q z + b^T z. \tag{27}$$

Now we use the gradient descent method with exact line search to optimize this problem, which means we have to find an optimal step size and direction for Eq. (2). In the gradient descent method, the direction $d_t$ is $-g_k$. The optimal step size $\gamma^*$ satisfies:

$$\gamma^* = \arg\min f(z_t + \gamma d_t)$$
$$= \arg\min f(z_t - \gamma g_t)$$
$$= \arg\min \left( \frac{1}{2}g_t^T Q g_t \gamma^2 - g_t^T g_t \gamma + f(z_t) \right). \tag{28}$$

Observing Eq. (28), which is a quadratic function of $\gamma$, we get the optimal $\gamma^*$:

$$\gamma^* = \frac{g_t^T g_t}{g_t^T Q g_t} \tag{29}$$

Next, We take the derivative of $h(x^*)$, and set the derivative to be 0, namely

$$Qz^* + b = 0 \tag{30}$$

$$z^* = -Q^{-1}b. \tag{31}$$

The optimization process can be written as Eq. (28) and the optimal step size $\gamma^*$ can be written as Eq. (29). First, we replace $f(z_k)$ and $f(z_{k+1})$ with Eq. (29) and Eq. (2).

$$f(z_{k+1}) = f(z_k - \gamma_k^* g_k) = \frac{1}{2}z_k^T Q z_k + b^T z_k - \frac{1}{2}\frac{(g_k^T g_k)}{g_k^T Q g_k} = f(z_k) - \frac{1}{2}\frac{(g_k^T g_k)^2}{g_k^T Q g_k}. \tag{32}$$

Firstly, we define $\kappa = cond(Q) = \frac{\lambda_{max}}{\lambda_{min}}$. Now, we can write the convergence rate of the problem (27) as follows:

$$\left| \frac{f(z_{k+1}) - f(z^*)}{f(z_k) - f(z^*)} \right| = \frac{\frac{1}{2}z_k^T Q z_k + b^T z_k - \frac{1}{2}\frac{(g_k^T g_k)^2}{g_k^T Q g_k} + \frac{1}{2}b^T Q^{-1}b}{\frac{1}{2}z_k^T Q z_k + b^T z_k + \frac{1}{2}b^T Q^{-1}b} \tag{33}$$

$$= 1 - \frac{(g_k^T g_k)^2}{(g_k^T Q g_k)(Qz_k + b)^T Q^{-1}(Qz_k + b)}. \tag{34}$$

Notice that from Eq. (33) to Eq. (A.1), we use the orthogonal property of $Q$, which is $Q^T = Q$. Then we plug Kantorovich inequality into Eq. (A.1) and obtain

$$1 - \frac{(g_k^T g_k)^2}{(g_k^T Q g_k)(Qz_k + b)^T Q^{-1}(Qz_k + b)} \leq 1 - \frac{4\lambda_{max}\lambda_{min}}{(\lambda_{max}+\lambda_{min})^2} \tag{35}$$

$$\leq \left(\frac{\lambda_{max}-\lambda_{min}}{\lambda_{max}+\lambda_{min}}\right)^2 \tag{36}$$

$$\leq \left(1 - \frac{2}{1+\kappa}\right)^2. \tag{37}$$

Through Eq. (A.1), we know that the convergence rate of the problem (27) using gradient descent with exact line search method is no faster than a constant, which is related to the condition number of $Q$.

**Lemma 3.2** Let $G(V, E)$ be a graph with $n$ nodes (with self-loop). $X$ is the feature matrix of G. Let $A$ be a $n \times n$ matrix such that

$$\begin{cases} A_{ij} > 0 & \text{if } (v_i, v_j) \in E, \\ A_{ij} = 0 & \text{else}, \end{cases}$$

Supposed $X^* = A^k X$ is a positive and proper $k$-step activation of graph G.

**Proof:** Firstly, we prove a lemma for support, which is that $(A^k)_{ij} \geq 0$ if the $i_{th}$ node and $j_{th}$ of G are $k$-step neighbours to each other, else $(A^k)_{ij} = 0$. We prove this by induction.

**Base case:** In the case K = 1, the lemma is true as the definition of A.

**Induction:** we assume the lemma is true for K then we prove it's also true for $K+1$ case.

$$A^{k+1} = A^K * A \tag{38}$$

Based on Eq. (38), we can see that

$$(A^{k+1})_{ij} = \sum_{k=1}^{n} (A^k)_{ij} A_{rj}, \tag{39}$$

which means that $A_{ij}^{k+1}$ is positive iff there exists some $r$ making $(A^k)_{ir}$ and $A_{rj}$ positive, ($r = 1, 2, ..., n$). Moreover, as our assumption, $(A^k)_{ir}$ is positive iff $i_{th}$ node and $r_{th}$ node of $G$ are $k$-step neighbors. Also, $A_{rj}$ is positive iff $r_{th}$ node and $j_{th}$ node of $G$ are 1-step neighbors as the definition of $A$. We call these three nodes $i$, $j$ and $r$ for short.

Thus we can get the conclusion that $(A^{k+1})_{ij}$ is positive iff there is a path between $i$ and $j$ going through $r$, whose length is $k + 1$. Considering that any $r$ ranging from $i$ to $n$ can be in the case so

that the condition of $(A^{k+1})_{ij}$'s positivity is equivalent to that $i$ and $j$ are $k+1$-step neighbor to each other. Now the lemma has been proved.

Then we can replace $X^* = A^k X$

$$X^*_{i:} = (A^k)_{i1} x_1^T + (A^k)_{i2} x_2^T + ... + (A^k)_{in} x_n^T. \tag{40}$$

According to the lemma we just proved, Eq. (40) represents a positive $k$-step activation of node $i$. Also, we add a self-loop on the graph so that any node is itself's $k$-step neighbor so that $(A^k)_{ii} 0$, then this activation is proper. Notice the node is arbitrarily chosen, so every node satisfies the properties above, then $X^* = A^k X$ is a positive and proper $k$-step activation of graph $G$. Our proof ends.

**Lemma 3.3.** The sum of finite positive and proper activations is still positive and proper.

*Proof:* It can be easily proved because the sum of finite non-negative real numbers is still non-negative.

**Theorem 3.3.** Let $L$ be the Laplacian matrix of graph $G$ and $I$ be the Identity matrix, then $\sum_{j=0}^{k} \alpha_j (2I - L)^j X, \forall \alpha_j \geq 0$, is a proper and positive activation of $G$.

*Proof:* It can be proved that $(2I - L)^j X$ is a proper and positive activation because of $(2I - L)$ have the same form with matrix A in Lemma2.2. And $\sum_{j=0}^{k} \alpha_j (2I - L)^j X$ meets Lemma 2.3. So $\sum_{j=0}^{k} \alpha_j (2I - L)^j X$ is a proper and positive activation of G. Our proof ends.

## A.2 DETAILS OF EXPERIMENTS

Due to the problems related to the version of the package, especially for pyg, and compatibility of datasets, we have to use two code structures to process our data and thus conduct training.

**Model Setup.** For Chameleon,Actor and Penn94, we followed the (Guo & Wei, 2023) structure. For both methods, we initialize the weight parameter to all 1s. The hidden size of the first MLP layers $h$ is set as 64, which is also the number of filter channels. And we tune all the parameters with Adam optimizer. For other datasets, we used the (He et al., 2021) structure. For FixMono, the parameter decoupling method is used for parameter initialization. For SimpleNet, we initialize the weight parameter to all 1s. For both models, we use a 2-layer MLP with 64 hidden units and we optimize the learning rate for the linear layer and the propagation layer. We also used dropout in the convolutional layer and the linear layer.

**Hyperparameter tuning.** We choose hyperparameters on the validation sets. We select hyperparameters from the range below with a maximum of 100 complete trials. For Chameleon, Actor and Penn94,

- Truncated Order polynomial series: $K, K1, K2 \in \{0, 1, 2, 3, 4, 5, 6\}$ ;
- Learning rates for linear layer: $\{ 0.008, 0.01, 0.012, 0.014, 0.02, 0.03, 0.04\}$ ;
- Learning rates for propagation layer: $\{ 0.01, 0.012, 0.014, 0.016, 0.0175, 0.02, 0.03, 0.04, 0.05\}$ ;
- Weight decays: $\{1e\text{-}8, \cdots, 1e\text{-}3\}$;
- Dropout rates: $\{0., 0.1, \cdots, 0.9\}$;

For other datasets,

- Truncated Order polynomial series: $K, K1, K2 \in \{0, 1, 2, 3, 4, 5, 6\}$ ;
- Learning rates for linear layer: $\{ 0.01, 0.012, 0.015, 0.02, 0.022, 0.025, 0.03, 0.04, 0.05\}$ ;
- Learning rates for propagation layer: $\{0.01, 0.02, 0.022, 0.024, 0.03, 0.04, 0.05 \}$ ;
- Weight decays: $\{0, 0.005\}$;
- Dropout rate: $\{0., 0.1, 0.2, 0.3\}$

## A.3 COMPARISON OF RUNNING TIME

The comparison of running time for all the methods is shown in Figure 4-7. As can be seen, the area under the curve is the total running time of each dataset. Obviously, SimpleNet's area is the smallest among other models which means that our model has extremely low computational overhead. Although the number of SimpleNet's total running epochs is a bit high, its running time per epoch is really short. It really verifies the convergence rate analysis of GNN since our model's polynomial may not have a faster convergence rate but the computational cost of the polynomial itself is so low that it can compensate for the lack of convergence speed.

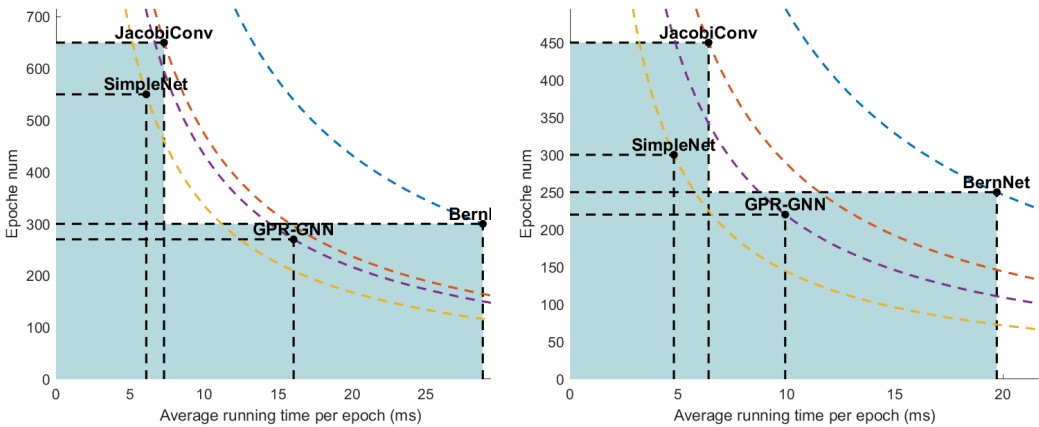

Figure 4: Running time on computer dataset.    Figure 5: Running time on cora dataset.

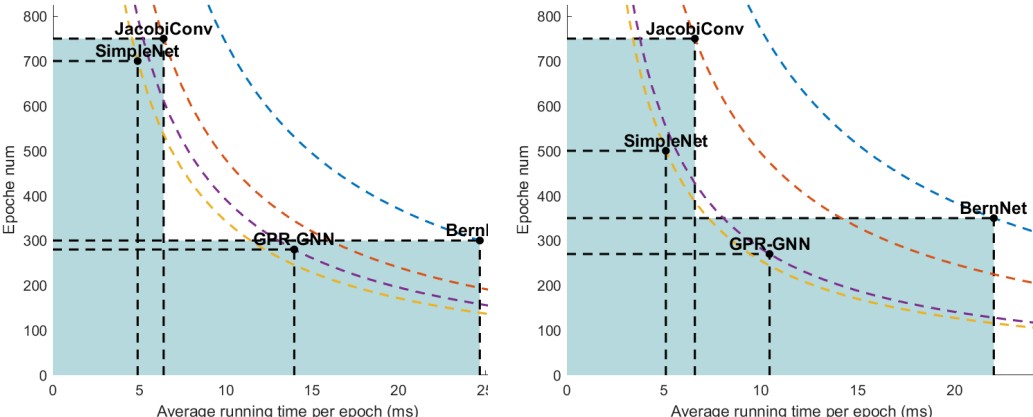

Figure 6: Running time on photo dataset.    Figure 7: Running time on PubMed dataset.

