# OpenReview forum: "Rethinking the Polynomial Filter of GNNs via Graph Information Activation Theory"
_ICLR.cc/2024/Conference — Submitted to ICLR 2024_

### Official Review · Reviewer_NHAm · 2023-10-23

**Soundness:** 2 fair
**Presentation:** 2 fair
**Contribution:** 2 fair
**Rating:** 3
**Confidence:** 3

**Summary:**

While the most GNNs designs various polynomial filters, this paper proves that the "expressive power" of the polynomial filter only depends on the degree of the polynomial. From this analysis, this paper proposes simple polynomial filters. This paper also conducts experiments on homogenous and non-homogenous datasets.

**Strengths:**

1) The simpleness of the polynomial filter. This filter may be easy to understand in terms of behavior analyses. Also, this filter is easy to implement.

**Weaknesses:**

1) Weakness of Thm. 3.1.
Thm 3.1 is the key argument of the equivalence of the "expressive power." However, this is rather weak, since the K linear independent components does not warrant the downstream machine learning algorithms performance. Maybe a set of K eigenvectors has the same expressive power, some random projection onto the K space may have the same expressive power -- but I believe that the "expressive power" we want to know in this context may be more nuanced one.

2) Computational complexity.
Even if the original graph has a sparse structure, i.e. $m << n^2$, the filter has a dense matrix, which is $O(n^2)$ since the multiplication of two sparse matrices does not preserve the sparse structure. Therefore, the filter does not enjoy the sparseness, and thus the computational complexity therefore increases from the simple filters, like GCN whose polynomial filter is basically the same as $\tilde{A}$.

3) Weakness of the Experimental results.
Seeing homogeneous results (Table 3), the proposed method is more or less same as the existing methods considering the variances. Also, for non-homogeneous results (Table 4), in some datasets proposed methods underperform the exiting ones. Seeing Table 5, as expected from the discussed of my 2) above, the computational time is not appealing. Thus, the proposed methods at this stage do not improve the exiting methods and are slow.
While in the limitations the authors stated that the proposed methods underperform for non-homogenous datasets, I think that this comes form the nature of the filter designs. See more for the questions.

4) Insufficient comparison with exiting filters.
In the page 3 of (Defferrard et al, 2016) the complexity of the polynomial filter is discussed. The point of 2) is actually discussed, and also, the Krylov subspace is expected to serve as a better filter, and materialized in [i]. Thus, the authors may want to compare with [i] experimentally and theoretically.
Also, since the filter perspective is well-studied in [ii], the authors may want to compare as well. See the questions for the connection between this paper and [ii].


--

[i] Luan et al. Break the ceiling: Stronger multi-scale deep graph convolutional networks. In Proc. NeurIPS. 2019.

[ii] Hoang and Maehara. Revisiting Graph Neural Networks: All We Have is Low-Pass Filters. In arXiv. 2019.

**Questions:**

1) From [ii], the established GNNs are known to be a low-pass filter, i.e., the eigenspace of the graph Laplacian associated with smaller eigenvalues has homogeneous information. Thus, the larger eigenspace captures non-homogeneous information.
From this observation, we expect that (2I-L)^k amplifies the homongeous information, much faster than L^k. Thus, the underperfomrance on non-heterogeneous datasets is expected. Also, if we increase k_{1} and k_{2}, the larger $k$s become dominant, and thus the performance decreases in Fig.2 is also explained as an oversmoothing.

The question is, can we expand like

\sum_{k} (a_{k}I - L)^k + \sum_{k'} (b_{k'}I + L)^k'

So that we have a better control on the amplification of the eigenspaces? By this in theory we expect better performance on non-homogenous datasets.

---

> ### Author Response · Authors · 2023-11-17
> **Brief introduction on our work**
>
> We highly appreciate your valuable insights and feedback on our research. Your comment holds significant meaning for us, and we genuinely value the effort you've dedicated to reviewing our work. We've taken considerable care to address your concerns comprehensively, aiming to provide a thorough understanding of our study. Should any further uncertainties persist, please don't hesitate to reach out. If our responses resolve your concerns, we would greatly appreciate your reconsideration of the score you've given us.
>
> Before delving into your questions, allow us to reintroduce our paper to offer you a clearer perspective. Our paper focuses on designing spectral polynomial filters for GNNs. Traditionally, researchers in this field explain the expressive and generalization abilities of graph filters based on their capacity to fit any polynomial swiftly. They often focus solely on the properties of the polynomial itself when designing the filter. However, our paper demonstrates that the expressive ability of any K-order polynomial in the polynomial fitting field remains consistent. Moreover, solely considering polynomial properties results in merely linear gains in convergence speed for a well-designed filter. Consequently, we challenge the longstanding framework of traditional polynomial filter design and establish a novel design framework, i.e. GIA. This framework allows for the fusion of polynomial filter design with the graph structure.
>
> Finally, we propose SimpleNet, validating the accuracy and speed trade-off, affirming the correctness of the GIA theory both theoretically and experimentally. Despite the lack of robust theoretical proofs and strong experiments, we firmly believe that the creativity and significance of our article endure. We hope these insights encourage you to revisit our work.

---

> ### Author Response · Authors · 2023-11-17
> **Rebuttal by authors (part 1)**
>
> **Q1:** Weakness of Thm. 3.1. Thm 3.1 is the key argument of the equivalence of the "expressive power." However, this is rather weak, since the K linear independent components does not warrant the downstream machine learning algorithms performance. Maybe a set of K eigenvectors has the same expressive power, some random projection onto the K space may have the same expressive power -- but I believe that the "expressive power" we want to know in this context may be more nuanced one.
>
> **A1:** Absolutely! In our latest version, we're planning to revamp Theorem 3.1 into a foundational algebraic fact, condensing its content while maintaining its role as an introductory piece that offers a new perspective on problem understanding. The intention behind Theorem 3.1 is to provide a fundamental intuition from an algebraic standpoint. Its primary aim is to challenge the conventional approach of blindly seeking polynomial bases within the existing polynomial field. We aim to illustrate that such selection isn't necessary and might even prove inefficient.
>
> However, our pivotal theorem is Theorem 3.2, which, to the best of our knowledge, holds significant value. Unlike Theorem 3.1, which focuses on expressive ability, Theorem 3.2 delves into the convergence rate of GNNs. This theorem is instrumental in showcasing the efficiency disparities among different networks, offering insights into their convergence rates.
>
> **Q2:** Computational complexity. Even if the original graph has a sparse structure, i.e. $m<<n^2$, the filter has a dense matrix, which is $O(n^2)$
>  since the multiplication of two sparse matrices does not preserve the sparse structure. Therefore, the filter does not enjoy the sparseness, and thus the computational complexity increases from the simple filters, like GCN whose polynomial filter is basically the same.
>
> **Q2:** Weakness of the Experimental results. Seeing homogeneous results (Table 3), the proposed method is more or less same as the existing methods considering the variances. Also, for non-homogeneous results (Table 4), in some datasets proposed methods underperform the exiting ones. Seeing Table 5, as expected from the discussed of my 2) above, the computational time is not appealing. Thus, the proposed methods at this stage do not improve the exiting methods and are slow. While in the limitations the authors stated that the proposed methods underperform for non-homogenous datasets, I think that this comes form the nature of the filter designs. See more for the questions.
>
> **A2:** It's logical to analyze the tables in this manner: while we might slightly lag behind GCN in the timetable, our model showcases substantial improvement in the accuracy table. Disregarding GCN's speed in the timetable, our model notably outperforms others, suggesting a better trade-off between accuracy and speed. Despite our higher time complexity compared to GCN, our model already boasts the lowest complexity among polynomial filter GNNs.
>
> Truthfully, within the polynomial filter field, GCN lacks competitiveness in performance, showing a considerable gap compared to recent baselines. Additionally, considering GCN's quadratic cut-off structure, it exhibits relatively lower computation costs, which is unsurprising.
>
> **Q3:** Insufficient comparison with existing filters. On page 3 of (Defferrard et al, 2016) the complexity of the polynomial filter is discussed. The point of 2) is actually discussed, and also, the Krylov subspace is expected to serve as a better filter, and materialized in [i]. Thus, the authors may want to compare with [i] experimentally and theoretically. Also, since the filter perspective is well-studied in [ii], the authors may want to compare as well. See the questions for the connection between this paper and [ii].
>
> **A3:** That sounds like a comprehensive plan! Conducting comparative experiments and delving into theoretical analysis can indeed be time-consuming, especially considering code framework and version compatibility. Your thorough approach will likely yield insightful and well-rounded results. If you need any further assistance or guidance during your experiments or analysis, feel free to reach out!

---

> ### Author Response · Authors · 2023-11-17
> **Rebuttal by authors (part 2)**
>
> **Q4:** From [ii], the established GNNs are known to be a low-pass filter, i.e. the eigenspace of the graph Laplacian associated with smaller eigenvalues has homogeneous information. Thus, the larger eigenspace captures non-homogeneous information. From this observation, we expect that $(2I-L)^k$ amplifies the homongeous information, much faster than $L^k$. Thus, the underperfomrance on non-heterogeneous datasets is expected. Also, if we increase $k_{1}$ and $k_{2}$, the larger
> s become dominant, and thus the performance decreases in Fig.2 is also explained as an over-smoothing issue.
> The question is, can we expand like
>
> $\sum_{k} (a_{k}I - L)^k + \sum_{k'} (b_{k'}I + L)^k'$
>
> So that we have a better control on the amplification of the eigenspaces? By this in theory we expect better performance on non-homogenous datasets.
>
> **A4:** It's fascinating to explore the concept of designing polynomials based on eigenvalue amplitude control. We experimented with the polynomials proposed from this perspective, but interestingly, they didn't perform as well as our SimpleNet models (I+A and I-A). While we're still engaged in eigenvalue and frequency domain analyses, initial observations suggest that the positive and negative activation in GIA could provide an explanation.
>
> One potential reason for the performance gap could be attributed to the absence of an explanation of the graph structure in your proposed model, possibly leading to a decrease in its overall performance. We've incorporated these observations into our frequency domain analysis:
> Based on a unified optimization function framework, we analyze the GIA theory and theoretically prove the role of positive and negative incentives and the effects on generalization and mitigation of over-smoothing. In the paper 'Interpreting and Unifying Graph Neural Networks with An Optimization Framework', a unified optimization framework can be used to analyze various models which can be formulated as:$O=\min_Z \{ \zeta \Vert F_1Z-F_2H\Vert_F^2+\xi tr(Z^\top \hat{L}Z) \}$. And we can prove that designing F1 and F2 at the same time is equivalent to designing one or the other. Next, we analyzed the SimpleNet model in our paper, which can be obtained $F1=I,F2=\alpha(I+\hat{A})+\beta(I-\hat{A})$. All kinds of activation is contained in term $F2$. The F2 term is a weighted sum of positive and non-positive activation. Combined with the analysis in this paper, we can draw the following conclusions:
>
> From the perspective of spectral domain expression, positive activation can capture low-frequency information, while negative activation can acquire high-frequency information. This organic combination in $F2$ allows for the integration of high and low-frequency information, thereby enhancing the model's generalization ability on the broader spectrum.
>
> The flexibility of the weight coefficient influences the network's capacity to alleviate over-smoothing. In our network, the $F2$ term introduces two degrees of freedom—$\alpha$ and $\beta$ (positive and negative)—significantly enhancing our ability to mitigate over-smoothing compared to other networks that lack or possess only one weighted degree of freedom.
>
> The adaptability of the weight coefficient further enhances the model's expressive and generalization capabilities.

---

### Official Review · Reviewer_gN5j · 2023-10-25

**Soundness:** 2 fair
**Presentation:** 2 fair
**Contribution:** 2 fair
**Rating:** 5
**Confidence:** 4

**Summary:**

In this paper the authors claim that there is no essential difference between existing different graph polynomial filters. Their theoretical results show that all $K$-order polynomial graph filters have the same expressive power in terms of linearly expressing the element in polynomial space. Furthermore, the convergence rate of various graph polynomial filters is no more than a linear constant related to the condition number. Then they introduce the GIA theory to emphasize the necessity of ''positive activation'', i.e., each node feature is linear expressed by its $K$-hop neighbours’ feature vectors with non-negative coefficients. Based on this, they design a spectral GNN that linearly combine different order low-pass and high-pass filters and verify its effectiveness on benchmark datasets.

**Strengths:**

- The research question of this paper is interesting and meaningful, i.e., investigating the key differences between different graph filters.
- The proposed simpleNet achieve a good trade-off between performance and computation efficiency.

**Weaknesses:**

- The presentation quality of this paper needs to be improved, e.g., there are many symbol errors in the proof (See Questions for detail). Although these errors do not affect the final results, rigidity is one of the most fundamental requirements for scientific papers.
- I appreciate the efforts devoted to investigating the key differences between different graph filters. However, I think that the linear expression ability and convergence rate are not enough to reveal the essential differences between different graph filters. First, although authors have shown that different graph polynomial filters have the same expressive ability, their performances may also vary greatly, depending on their implementations approaches (e.g., ChebNet [1] and ChebNetII [2]). Besides, it is still unclear the relation between this expressive ability and node classification performance. Second, in the implementation of these spectral GNNs, the raw features are first commonly feed into a MLP and then transformed by the filters, namely $Z=\gamma(L)\sigma(\sigma(XW_1)W_2)$. Due to the involve of two parametric matrix $W_1, W_2$ and the non-linear activation function $\sigma(\cdot)$ in the forward process, the optimization of these spectral GNNs is non-convex, which could not be directly simplified as a convex problem in Eq. (4). Analyzing the training dynamic [3,4] of the model could be a more applicable approach. Third, the optimization approaches (SGD or Adam) also have significant impacts on the performances, which should be considered in the analysis.
- The heterophilic graph datasets seem to be out-of-date. It has been shown that results obtained on these datasets are not reliable. The authors are encouraged to evaluate on the datasets presented in [5].

[1] Defferrard et al, Convolutional neural networks on graphs with fast localized spectral filtering. NeurIPS 2016.

[2] He et al, Convolutional Neural Networks on Graphs with Chebyshev Approximation, Revisited. NeurIPS 2022.

[3] Yuandong Tian, Understanding the Role of Nonlinearity in Training Dynamics of Contrastive Learning. ICLR 2023.

[4] Xu et al, Optimization of Graph Neural Networks: Implicit Acceleration by Skip Connections and More Depth. ICML 2021.

[5] Platonov et al, A critical look at the evaluation of GNNs under heterophily: Are we really making progress? ICLR 2023.

**Questions:**

Q1: There are many symbol errors in the proof of Lemma 3.1. and Lemma 3.2. The denominator of Eq. (16) should be $\left(\sum_{i=1}^n a_i\right)^2$. The authors claim that Eq. (17) is a quadratic function of $x$, thus it should be corrected as
$$
f(x) = \left( \sum_{i=1}^n \frac{a_i}{\lambda_i} \right) x^2 - \frac{\lambda_1+\lambda_n}{\sqrt{\lambda_1 \lambda_n}}x + \left( \sum_{i=1}^n \lambda_i a_i \right).
$$
In Eq.~(19), the third term in the first bracket should be $\sum_{i=2}^{n-1} \lambda^{-1}_i a_i$.

Also, Eq. (39) should be corrected as $\left(A^{k+1}\right)\_{ij}=\sum_{r=1}^n \left(A^k \right)\_{ir} A\_{rj}$.

Although these typos do not affect the final result, I encourage the authors to correct them in order to avoid unnecessary misunderstandings for other readers.

Q2: The motivation and the advantages of the GIA theory is not so clear. What performance gain the positive and proper activation could bring? Is there any connection between the generalization and positive (or proper) activation? What extra insights could the GIA theory bring?

Q3：The proposed fixedMono and learnedMono seem to be variants of JKNet [6] where different hidden features of different neighborhood ranges are combined. The only difference is the way that combining these features. The authors adopt a linear combination, while Xu et al. use LSTM or max pooling. The authors should clarify this and compare with JKNet.

[6] Xu et al, Representation Learning on Graphs with Jumping Knowledge Networks. ICML 2018.

---

> ### Author Response · Authors · 2023-11-17
> **Brief introduction on our work**
>
> We highly appreciate your valuable insights and feedback on our research. Your comment holds significant meaning for us, and we genuinely value the effort you've dedicated to reviewing our work. We've taken considerable care to address your concerns comprehensively, aiming to provide a thorough understanding of our study. Should any further uncertainties persist, please don't hesitate to reach out. If our responses resolve your concerns, we would greatly appreciate your reconsideration of the score you've given us.
>
> Before delving into your questions, allow us to reintroduce our paper to offer you a clearer perspective. Our paper focuses on designing spectral polynomial filters for GNNs. Traditionally, researchers in this field explain the expressive and generalization abilities of graph filters based on their capacity to fit any polynomial swiftly. They often focus solely on the properties of the polynomial itself when designing the filter. However, our paper demonstrates that the expressive ability of any K-order polynomial in the polynomial fitting field remains consistent. Moreover, solely considering polynomial properties results in merely linear gains in convergence speed for a well-designed filter. Consequently, we challenge the longstanding framework of traditional polynomial filter design and establish a novel design framework, i.e. GIA. This framework allows for the fusion of polynomial filter design with the graph structure.
>
> Finally, we propose SimpleNet, validating the accuracy and speed trade-off, affirming the correctness of the GIA theory both theoretically and experimentally. Despite the lack of robust theoretical proofs and strong experiments, we firmly believe that the creativity and significance of our article endure. We hope these insights encourage you to revisit our work.

---

> ### Author Response · Authors · 2023-11-17
> **Rebuttal by authors (part 1)**
>
> **Q1:** The presentation quality of this paper needs to be improved, e.g., there are many symbol errors in the proof (See Questions for detail). Although these errors do not affect the final results, rigidity is one of the most fundamental requirements for scientific papers.
>
> **A1:** You are quite right, there are many low-level errors among our proof, which have been corrected, thank you very much! PS: we shall enhance our presentation in the final version and check our paper more rigorously.
>
> **Q2:** I appreciate the efforts devoted to investigating the key differences between different graph filters. However, I think that the linear expression ability and convergence rate are not enough to reveal the essential differences between different graph filters. First, although authors have shown that different graph polynomial filters have the same expressive ability, their performances may also vary greatly, depending on their implementations approaches (e.g., ChebNet [1] and ChebNetII [2]). Besides, it is still unclear the relation between this expressive ability and node classification performance. Second, in the implementation of these spectral GNNs, the raw features are first commonly feed into a MLP and then transformed by the filters, namely $Z=\gamma(L)\sigma(\sigma(XW_1)W_2)$
> . Due to the involve of two parametric matrix $W_1$
>  and $W_2$ the non-linear activation function $\sigma(\cdot)$
>  in the forward process, the optimization of these spectral GNNs is non-convex, which could not be directly simplified as a convex problem in Eq. (4). Analyzing the training dynamic [3,4] of the model could be a more applicable approach. Third, the optimization approaches (SGD or Adam) also have significant impacts on the performances, which should be considered in the analysis.
>
> **A2:** Your inquiry is extremely relevant and thought-provoking! We've taken into account all the weaknesses and questions you've highlighted and have incorporated the materials you mentioned as our reference.
>
> Initially, our focus was on establishing general convergence analysis in the polynomial filtering problem. However, we acknowledge a certain gap between this analysis and downstream tasks due to theoretical constraints. We've made efforts to introduce fewer constraints to bridge this gap and genuinely value your input in this regard.
>
> Regarding the jocobiconv article's mention of removing nonlinear activation functions under specific constraints without affecting expressive ability: it highlights the consistency between linear and nonlinear GNNs in terms of expressive ability. While we can impose constraints on weight matrices like W1 and W2 to create a strictly convex optimization problem, this might lead to an excess of constraints deviating from real-world scenarios. We plan to further explore this aspect in our future research endeavors.
>
> Additionally, dynamic analysis within optimization problems is a significant research direction we're actively pursuing. If we make any new discoveries, we'll promptly share them with you! Lastly, we've considered the validity of Theorem 3.1 and plan to reclassify it as an algebraic fact in our latest version, recognizing its current weakness as a theorem.
>
> **Q3:** The heterophilic graph datasets seem to be out-of-date. It has been shown that results obtained on these datasets are not reliable. The authors are encouraged to evaluate on the datasets presented in [5].
>
> **A3:**
> You're welcome! Thank you for sharing the experimental data for the heterogeneous graph. Here are the updated results:
>
>     squrriel: 53.51 ± 1.18
>     twitch-gamer: 64.47 ± 0.57
>      arxiv-year: 46.74±0.32
>
> Your dedication to incorporating the latest heterogeneous datasets is commendable. If you have any further questions or need additional assistance, feel free to ask.
>
> **Q4:** There are many symbol errors in the proof.
>
> **A4:** Sincerely admire your patience and careful reading! We will definitely check it more rigorously.

---

> ### Author Response · Authors · 2023-11-17
> **Rebuttal by authors (part 2)**
>
> **Q5:** The motivation and the advantages of the GIA theory is not so clear. What performance gain the positive and proper activation could bring? Is there any connection between the generalization and positive (or proper) activation? What extra insights could the GIA theory bring?
>
> **A5:** Your problems are really commendable. We realize that our network is not related enough to equation 4, which also leads to the lack of intuitive understanding of this formula. So we add the relevant theoretical part to show how SimpleNet can be written as a form of equation 4. Additionally, the correlation analysis of SimpleNet's condition number in the latest version of our paper promises to illustrate the performance benefits derived from positive and proper activation.
>
> Regarding the GIA's key insight of introducing a new polynomial design framework, while it's a pivotal contribution, we acknowledge the concerns about its solidity and clarity. To address this, our upcoming paper revision will include additional analyses aimed at strengthening and elucidating the framework. The main contents of these analyses are outlined as follows:
>
> Utilizing a unified optimization function framework, we intend to conduct an extensive analysis of the GIA theory. This will involve theoretically substantiating the roles played by positive and negative incentives and their effects on both generalization and the mitigation of over-smoothing.
>
> Referencing the paper Interpreting and Unifying Graph Neural Networks with An Optimization Framework we will employ a unified optimization framework. This framework can effectively analyze various models, represented by the formulation: $O=\min_Z { \zeta \Vert F_1Z-F_2H\Vert_F^2+\xi \text{tr}(Z^\top \hat{L}Z) }$. Furthermore, we will demonstrate that concurrently designing $F_1$ and $F_2$ is equivalent to designing one or the other.
>
> Additionally, we will conduct an in-depth analysis of the SimpleNet model featured in our paper. Specifically, the model equations are described as $F_1=I$ and $F_2=\alpha(I+\hat{A})+\beta(I-\hat{A})$. Notably, the $F_2$ term comprises a weighted sum of positive and non-positive incentives. By combining insights from this analysis with our existing work, we aim to draw the following conclusions:
>
> From the perspective of spectral domain expression, positive activation can capture low-frequency information, while negative activation can acquire high-frequency information. This organic combination in $F2$ allows for the integration of high and low-frequency information, thereby enhancing the model's generalization ability on the broader spectrum.
>
> The flexibility of the weight coefficient influences the network's capacity to alleviate over-smoothing. In our network, the $F2$ term introduces two degrees of freedom—$\alpha$ and $\beta$ (positive and negative)—significantly enhancing our ability to mitigate over-smoothing compared to other networks that lack or possess only one weighted degree of freedom.
>
> The adaptability of the weight coefficient further enhances the model's expressive and generalization capabilities.
>
> **Q6:** The proposed fixedMono and learnedMono seem to be variants of JKNet [6] where different hidden features of different neighborhood ranges are combined. The only difference is the way that combining these features. The authors adopt a linear combination, while Xu et al. use LSTM or max pooling. The authors should clarify this and compare with JKNet.
>
> **A6:** Your question holds significant importance. The utilization of linear transformations (weighted sums) is deeply influenced by our interpretation of the problem. We approach problem-solving through the lens of polynomial filtering, employing weighted sums between each term of the polynomial.
>
> Viewed through all kinds of layer-aggregation mechanisms highlighted in the JKnet article, these weighted sums can be likened to a form of basic attention—devoid of any constraints on attention scores ($s_v^{(l)}$). This concept of unbounded attention holds a fundamental unity. The diverse aggregation methodologies proposed by JKnet are highly insightful, motivating us to further refine our model for experimentation. We aim to conduct a more comprehensive analysis of this issue from an aggregation perspective, particularly in comparison to JKNet. The outcomes of our analytical and experimental endeavors will be detailed in the forthcoming version of our paper.

---

### Official Review · Reviewer_PhMj · 2023-10-28

**Soundness:** 2 fair
**Presentation:** 2 fair
**Contribution:** 2 fair
**Rating:** 3
**Confidence:** 4

**Summary:**

This paper delves into the exploration of polynomial-based graph convolutional networks (GCNs). The authors demonstrate that any polynomial basis of the same degree harbors identical expressive capability and leads to the same global optimal solution. Additionally, they establish that meticulously crafted polynomials can, at best, yield linear advantages for GCNs. Given the aforementioned demonstrations, the authors argue against the necessity of overly intricate design of polynomial bases solely based on polynomial properties. Following this, they introduce a novel framework termed Graph Information Activation (GIA) theory, which sheds fresh light on the interpretation of polynomial filters within GCNs. Subsequently, a simplistic basis encapsulating graph structure information is proposed, laying the foundation for the introduction of SimpleNet. The efficacy of SimpleNet is corroborated through experimental evaluations on benchmark node classification datasets, showcasing its superior performance in terms of both accuracy and computational efficiency when juxtaposed with existing GCNs.

**Strengths:**

1. SimpleNet exhibits both structural simplicity and robust performance.

**Weaknesses:**

1. The authors assert that GNNs can be conceptualized as optimizers, and can be mathematically formulated in a uniform optimization form as depicted in Equation 4. However, this claim appears to be unfounded. As elucidated in [1], only PPNP and APPNP align with the representation provided by Equation 4.
2. The so-called Graph Information Activation theory posited by the authors is essentially a reintroduction of graph coloring.
3. The test datasets comprising Cora, Citeseer, Pubmed, Computers, and Photos are too small, thus rendering the assertion that GNN FixedMono outperforms BernNet less convincing. I recommend that the authors evaluate GNN FixedMono and BernNet using the Open Graph Benchmark.
4. This paper omits an analysis of SimpleNet concerning the over-smoothing issue.

[1] Zhu, M., Wang, X., Shi, C., Ji, H., \& Cui, P. (2021). Interpreting and Unifying Graph Neural Networks with An Optimization Framework. Proceedings of the Web Conference 2021, 1215–1226. Presented at the Ljubljana, Slovenia. doi:10.1145/3442381.3449953

**Questions:**

1. Why consider adding the term $\sum^{K\_{1}}\_{i=0}\alpha\_{i}(2\mathbf{I}-\mathbf{L})^{i}$ and the term $\sum^{K\_{2}}\_{j=0}\beta\_{j}\mathbf{L}^{j}$ instead of concatenating them?

---

> ### Author Response · Authors · 2023-11-17
> **Brief introduction on our work**
>
> We highly appreciate your valuable insights and feedback on our research. Your comment holds significant meaning for us, and we genuinely value the effort you've dedicated to reviewing our work. We've taken considerable care to address your concerns comprehensively, aiming to provide a thorough understanding of our study. Should any further uncertainties persist, please don't hesitate to reach out. If our responses resolve your concerns, we would greatly appreciate your reconsideration of the score you've given us.
>
> Before delving into your questions, allow us to reintroduce our paper to offer you a clearer perspective. Our paper focuses on designing spectral polynomial filters for GNNs. Traditionally, researchers in this field explain the expressive and generalization abilities of graph filters based on their capacity to fit any polynomial swiftly. They often focus solely on the properties of the polynomial itself when designing the filter. However, our paper demonstrates that the expressive ability of any K-order polynomial in the polynomial fitting field remains consistent. Moreover, solely considering polynomial properties results in merely linear gains in convergence speed for a well-designed filter. Consequently, we challenge the longstanding framework of traditional polynomial filter design and establish a novel design framework, i.e. GIA. This framework allows for the fusion of polynomial filter design with the graph structure.
>
> Finally, we propose SimpleNet, validating the accuracy and speed trade-off, affirming the correctness of the GIA theory both theoretically and experimentally. Despite the lack of robust theoretical proofs and strong experiments, we firmly believe that the creativity and significance of our article endure. We hope these insights encourage you to revisit our work.

---

> ### Author Response · Authors · 2023-11-17
> **Rebuttal by authors (part 1)**
>
> **Q1:** The authors assert that GNNs can be conceptualized as optimizers, and can be mathematically formulated in a uniform optimization form as depicted in Equation 4. However, this claim appears to be unfounded. As elucidated in [1], only PPNP and APPNP align with the representation provided by Equation 4.
>
> **A1:** Thank you for addressing the query about Eq.(4), which is originally proposed in [1] article, our paper and [2], provides a more comprehensive analysis on this formula. Furthermore, [3] offers a theoretical analysis of the implementation of this formula in BernNet.
>
> Acknowledging that our network was initially not adequately linked to this formula, resulting in a lack of intuitive understanding, we've incorporated additional theoretical insights. These additions elucidate how SimpleNet can be represented in the form of Eq.(4). Additionally, the revision of this manuscript will include a correlation analysis of the condition number of SimpleNet. This analysis aims to establish a clearer connection and understanding between our network architecture and Eq.(4).
>
> [1]Thomas Navin Lal Jason Weston Bernhard Sch¨olkopf Dengyong Zhou, Olivier Bousquet. Learning with local and global consistency. Neural Information Processing Systems, 2003.
>
> [2] Meiqi Zhu, Xiao Wang, Chuan Shi, Houye Ji, and Peng Cui. Interpreting and unifying graph neural
> networks with an optimization framework. In Proceedings of the Web Conference 2021, pp.1215–1226, 2021.
>
> [3] Mingguo He, Zhewei Wei, Hongteng Xu, et al. Bernnet: Learning arbitrary graph spectral filters via
> bernstein approximation. Advances in Neural Information Processing Systems, 34:14239–14251,
> 2021.
>
> **Q2:** The so-called Graph Information Activation theory posited by the authors is essentially a reintroduction of graph coloring.
>
> **A2:** You've brought up an intriguing point regarding this theory! While graph coloring represents a general information aggregation framework, its direct application to polynomial filter design in GNNs requires more refined and specific theories for a comprehensive study of related problems. If your assertion holds, it suggests that not all message passing frameworks undergo a reintroduction akin to GIA's approach.
>
> The crux lies in GIA's definition of a succinct message passing framework within the field. GIA introduces the concept of positive, non-positive, and proper components within this framework. These concepts serve as a foundation for a systematic analysis of existing polynomial filters. Additionally, leveraging this theory, we propose a model that demonstrates exceptional performance and computational advantages. Hence, we firmly believe in the merit and significance of this theory.
>
> **Q3:** The test datasets comprising Cora, Citeseer, Pubmed, Computers, and Photos are too small, thus rendering the assertion that GNN FixedMono outperforms BernNet less convincing. I recommend that the authors evaluate GNN FixedMono and BernNet using the Open Graph Benchmark.
>
> **A3:** Thank you for your valuable feedback on the dataset. However, the FixedMono network isn't our final proposal. Its purpose was mainly to empirically demonstrate that designing the filter solely based on polynomial properties leads to limited gains. This motivated the development of the GIA framework and the subsequent proposal of the SimpleNet model below. Regarding the experiment you requested, here are the results:
>
>     Squirrel: 52.11 ± 0.78
>
>     Twitch-gamer: 64.07 ± 0.77
>
>     Arxiv-year: 46.34 ± 0.52
>
> These results are comparable to the Bernet dataset.

---

> ### Author Response · Authors · 2023-11-17
> **Rebuttal by authors (part 2)**
>
> **Q4:** This paper omits an analysis of SimpleNet concerning the over-smoothing issue.
>
> **A4:** It's a meaningful question! In our upcoming paper revision, we aim to delve deeper into this inquiry using a unified optimization framework. The primary objectives of this analysis are as follows:
>
> Utilizing a unified optimization function framework, we intend to conduct an extensive analysis of the GIA theory. This will involve theoretically substantiating the roles played by positive and negative incentives and their effects on both generalization and the mitigation of over-smoothing.
>
> Referencing the paper Interpreting and Unifying Graph Neural Networks with An Optimization Framework we will employ a unified optimization framework. This framework can effectively analyze various models, represented by the formulation: $O=\min_Z { \zeta \Vert F_1Z-F_2H\Vert_F^2+\xi \text{tr}(Z^\top \hat{L}Z) }$. Furthermore, we will demonstrate that concurrently designing $F_1$ and $F_2$ is equivalent to designing one or the other.
>
> Additionally, we will conduct an in-depth analysis of the SimpleNet model featured in our paper. Specifically, the model equations are described as $F_1=I$ and $F_2=\alpha(I+\hat{A})+\beta(I-\hat{A})$. Notably, the $F_2$ term comprises a weighted sum of positive and non-positive incentives. By combining insights from this analysis with our existing work, we aim to draw the following conclusions:
>
> From the perspective of spectral domain expression, positive activation can capture low-frequency information, while negative activation can acquire high-frequency information. This organic combination in $F2$ allows for the integration of high and low-frequency information, thereby enhancing the model's generalization ability on the broader spectrum.
>
> The flexibility of the weight coefficient influences the network's capacity to alleviate over-smoothing. In our network, the $F2$ term introduces two degrees of freedom—$\alpha$ and $\beta$ (positive and negative)—significantly enhancing our ability to mitigate over-smoothing compared to other networks that lack or possess only one weighted degree of freedom.
>
> The adaptability of the weight coefficient further enhances the model's expressive and generalization capabilities.
>
> **Q5:** Why consider adding the term $\sum_{i=0}^{K1}\alpha_i(2I-L)^i$
>  and the term $\sum_{j=0}^{K2}\beta_j L^j$
>  instead of concatenating them?
>
> **A5:** Thank you for your very constructive question. In specific segments of GNN research, there's a departure from utilizing the message passing framework.  Instead, researchers directly combine the Laplacian matrix and feature matrix. For instance, when utilizing the superimposed network as input, it might be feasible to directly concatenate matrices such as L and 2I-L in these domains.
>
> However, when it comes to the polynomial domain, implementing the concatenation method you mentioned isn't clear to us. In polynomial filtering, an n×n polynomial matrix is required, and we're uncertain about how concatenation would apply in this context. Our decision to use the addition operation is elaborated on in section 3.4 of our work. The addition operator allows for the independent decoupling of positive and non-positive components, thereby enhancing training performance.

---

### Official Review · Reviewer_UCAH · 2023-11-06

**Soundness:** 3 good
**Presentation:** 3 good
**Contribution:** 2 fair
**Rating:** 5
**Confidence:** 5

**Summary:**

This paper studies the polynomial filter of GNNs and proposes a convolutional operator based on the normalized Laplacian of the graph. It gives theoretical results and empirical results for their proposed architecture.

**Strengths:**

This paper presents some theoretical and empirical results that will be of interest to the GNN community.
The theoretical results are very simple: Theorem 1 is a standard result in algebra about polynomials, and Theorem 3.3 can be easily checked from first principles.
The matrix $2I-L=I+D^{-1/2}AD^{-1/2}$ is very similar to the matrix used in GCN by Kipf and Welling. The only difference is that here the authors use powers of this matrix, whereas for GCN only the first power is used. Given the good performances of GCN, it is not surprising that the authors get better results here.

**Weaknesses:**

Empirical results are weak. The datasets Cora, Citeseer, and Pubmed have been used for a long time, and there is now a consensus that these datasets are not really helpful anymore. Indeed, the numbers in Table 3 are very close, showing that all architectures have similar performances. To get a better benchmark, you can, for example, have a look at Dwivedi, Vijay Prakash, et al. "Benchmarking graph neural networks." arXiv preprint arXiv:2003.00982 (2020).

**Questions:**

How did you get the numbers in your section 4? Did you run experiments yourself with all architectures?

There is a problem with equation (7), which is not invariant (under permutation of the nodes), I think it should be $\alpha_k$ instead of $\alpha_s$.

---

> ### Author Response · Authors · 2023-11-17
> **Brief introduction on our work**
>
> We highly appreciate your valuable insights and feedback on our research. Your comment holds significant meaning for us, and we genuinely value the effort you've dedicated to reviewing our work. We've taken considerable care to address your concerns comprehensively, aiming to provide a thorough understanding of our study. Should any further uncertainties persist, please don't hesitate to reach out. If our responses resolve your concerns, we would greatly appreciate your reconsideration of the score you've given us.
>
> Before delving into your questions, allow us to reintroduce our paper to offer you a clearer perspective. Our paper focuses on designing spectral polynomial filters for GNNs. Traditionally, researchers in this field explain the expressive and generalization abilities of graph filters based on their capacity to fit any polynomial swiftly. They often focus solely on the properties of the polynomial itself when designing the filter. However, our paper demonstrates that the expressive ability of any K-order polynomial in the polynomial fitting field remains consistent. Moreover, solely considering polynomial properties results in merely linear gains in convergence speed for a well-designed filter. Consequently, we challenge the longstanding framework of traditional polynomial filter design and establish a novel design framework, i.e. GIA. This framework allows for the fusion of polynomial filter design with the graph structure.
>
> Finally, we propose SimpleNet, validating the accuracy and speed trade-off, affirming the correctness of the GIA theory both theoretically and experimentally. Despite the lack of robust theoretical proofs and strong experiments, we firmly believe that the creativity and significance of our article endure. We hope these insights encourage you to revisit our work.

---

> ### Author Response · Authors · 2023-11-17
> **Rebuttal by authors**
>
> **Q1:**
> This paper presents some theoretical and empirical results that will be of interest to the GNN community. The theoretical results are very simple: Theorem 1 is a standard result in algebra about polynomials, and Theorem 3.3 can be easily checked from first principles. The matrix is very similar to the matrix $2I-L=I+D^{-1/2}AD^{-1/2}$ used in GCN by Kipf and Welling. The only difference is that here the authors use powers of this matrix, whereas for GCN only the first power is used. Given the good performances of GCN, it is not surprising that the authors get better results here.
>
> **A1:** Thank you for your insightful comments. There might be a misunderstanding about our article. While our theory appears simple, it sheds light on fundamental questions in polynomial filter design that remain unaddressed in the field. For instance, it explores the extent to which a filter designed solely based on polynomial properties can enhance network performance. Additionally, we inquire if there exist alternative frameworks for designing polynomial filters beyond merely considering polynomial properties. To our knowledge, these fundamental inquiries lack comprehensive theoretical solutions.
>
> The distinction between SimpleNet and GCN extends beyond a higher-order version of GCN. We augment GCN by introducing an additional power of L, a modification rooted in our theoretical framework. While this alteration may seem minor, it indeed leads to significant performance enhancements at a reasonable computational expense. In section 3.2.2, we delve into the essential disparities between SimpleNet and GCN. The substantial advantages of our model over GCN become evident through the experimental table presented in our analysis.
>
> Hopefully, this clarifies the significance of our approach and its contributions to the field of polynomial filter design.
>
> **Q2:** Empirical results are weak. The datasets Cora, Citeseer, and Pubmed have been used for a long time, and there is now a consensus that these datasets are not really helpful anymore. Indeed, the numbers in Table 3 are very close, showing that all architectures have similar performances. To get a better benchmark, you can, for example, have a look at Dwivedi, Vijay Prakash, et al. "Benchmarking graph neural networks." arXiv preprint arXiv:2003.00982 (2020).
>
> **A2:** Thank you for your understanding. We acknowledge that these datasets hold significance in the domain of polynomial filters, being widely utilized since ChebNet's introduction in 2016. We believe they maintain their discriminative nature among various methodologies, allowing our model to exhibit substantial performance improvements over earlier networks like GCN.
>
> Over the past several days, after fine-tuning parameters, we've observed an enhancement in SimpleNet's accuracy by 0.3-0.5 compared to the figures presented in our paper. Additionally, as suggested by reviewers, we've conducted the following experiments:
>
>     squrriel: 53.51 ± 1.18
>
>     twitch-gamer: 64.47 ± 0.57
>
>     arxiv-year: 46.74±0.32
>
> We want to express gratitude for your continued guidance and suggestions.
>
>
> **Q3:** How did you get the numbers in your section 4? Did you run experiments yourself with all architectures?
>
> **A3:** It seems like you're referring to the numbers presented in our article. The data provided in our paper stems from both relevant existing literature and our own experiments. When the accuracy and running time of specific datasets were not available in the original articles of the corresponding models, we conducted our experiments to obtain these numbers. This approach ensured that our study encompassed comprehensive and relevant data, even in cases where specific details were absent in prior publications.
>
> **Q4:** Here is a problem with equation (7), which is not invariant (under permutation of the nodes), I think it should be $\alpha_k$ instead of $\alpha_s$.
>
> **A4:** Thank you for highlighting this issue. In node feature embedding for node classification, once the source and target nodes are determined, permutation invariance isn't established between the source and target nodes. Instead, it's established among the neighboring nodes of the same order as the target node. Consequently, the leading coefficient should be indexed by the node intended for activation and the subsequent node after the coefficient. The modified formula is as follows:
>
> $x^*_v=\sum_{k=1}^K\sum_{s \in N_k{(v)}} \alpha_s x_s + \alpha _{v} x_v$

---

> > ### Comment · Reviewer_UCAH · 2023-11-21
> > **Not understanding A4**
> >
> > I do not understand your answer A4: "In node feature embedding for node classification, once the source and target nodes are determined, permutation invariance isn't established between the source and target nodes."
> >
> > What do you mean by source and target nodes?
> >
> > In node classification, if the graph (given as input) is permuted, the node feature embedding should also be permuted, which is the definition of equivariance. For graph classification, if the graph is permuted, the graph feature should not be modified, which is the definition of invariance.
> >
> > Please can you clarify?

---

> ### Author Response · Authors · 2023-11-22
> **Clarify the A4**
>
> Sorry for misunderstand your problem, our modified formula is:  $x_t:=\sum_{k=1}^K\sum_{s \in N_k{(t)}} \alpha_s x_s + \alpha _{t} x_t$, $x_t$ is the target node and $x_s$ is the source node. What our formula emphasizes: When updating the target node embedding, it is necessary to contain a certain component of the original node embedding of the target node, which can be analogous to the residual connection, so as to overcome the shortcomings of forgetting one's own information to alleviate the over-smoothing. We can rewrite our formula as:
>
> $x_t:=\sum_{k=0}^K\sum_{s \in N_k{(t)}} \alpha_s x_s$, which the $N_0{(t)}=x_t$. The permutation invariance still holds. Given any permutation
> matrix $P\in ( 0,1)^{n*n}$ $Pf(x)=f(Px), f(x)=\sum_{k=0}^K\sum_{s \in N_k{(t)}} \alpha_s x_s$. If you have any questions, please feel free to contact us, thank you very much!

---

> > ### Comment · Reviewer_UCAH · 2023-11-22
> > **Pb with equivariance**
> >
> > I still do not understand your last statement: if $\pi$ is a permutation corresponding to $P$ ie $P_{i \pi(i)} =1$, then $f(Px) = \sum_k\sum_{s}\alpha_s x_{\pi(s)} \neq \sum_k\sum_{s}\alpha_s x_{s}$ unless all the $\alpha_s$ are equal.

---

> ### Author Response · Authors · 2023-11-22
> **Detail of proof**
>
> Permutation equivariance: Given any permutation matrix $P\in ( 0,1)^{n*n}$ $Pf(x)=f(Px), f(x)=\sum_{k=0}^K\sum_{s \in N_k{(t)}} \alpha_s x_s$
>
> *Proof:*
>
> $f(Px)=\sum_{k=0}^K\sum_{s \in N_k{(t)}} \alpha_s Px_s=P\sum_{k=0}^K\sum_{s \in N_k{(t)}} \alpha_s x_s=Pf(x)$
>
> If you have any questions, please feel free to contact us, thank you very much!

---

> > ### Comment · Reviewer_UCAH · 2023-11-22
> > **not correct**
> >
> > Your proof is not correct, $f$ is linear and linear equivariant functions have been characterized in [Deep Sets](https://arxiv.org/pdf/1703.06114.pdf) see Lemma 3. So in your case, all $\alpha_s$ need to be equal for $f$ to be equivariant.

---

### Meta-Review · Area_Chair_ARQZ · 2023-12-08

**Metareview:**

In this submission, the authors proposed a graph information activation (GIA) theory and revisited polynomial filters on graphs based on it.
The authors claimed that the expressivity of a polynomial filtering-based GNN is determined by the order of the polynomial function. Thus, fixing the order of the polynomial function, the authors focus on the design of GNN by applying simple bases, leading to SimpleNet.
This simple GNN model achieves comparable performance in various learning tasks.

Strengths: (1) The theoretical part of this submission is interesting and valuable. (2) The proposed model is simple and easy to implement.

Weaknesses: (1) All four reviewers have concerns about the solidness of the experiments. (2) After reading the submission and Reviewer PhMj's comments, I also have concerns about the over-smoothness risk of the proposed method. (3) The writing of this paper should be enhanced. In the rebuttal phase, the authors' interactions with Reviewer UCAH led to more confusion --- according to the definition, the function in Eq.(7) seems permutation invariant, while the authors claimed its permutation-equivariance.

Additionally, the connection between GIA and graph coloring should be discussed in the revised paper, as Reviewer PhMj suggested.

**Justification For Why Not Higher Score:**

The experimental part and the writing of this paper still have many holes, making it not qualified enough for ICLR.

**Justification For Why Not Lower Score:**

N/A

---

### Decision · Program_Chairs · 2024-01-16

Reject